# Chemical staining for fundamental studies and optimization of binders in Li-ion battery negative electrodes

Stanislaw P. Zankowski [1,2] ✉, Samuel Wheeler[1,2], Thomas Barthelay [1,2], Wai Man Chan [3], Michael Metzler [1] & Patrick S. Grant [1,2] ✉

The spatial distribution of binders in Li-ion battery electrodes is critical to electrode performance, yet remains challenging to visualise, limiting binder optimisation efforts to chemical modifications rather than spatial control. Here, we show an accessible approach to staining carboxymethyl cellulose and styrene butadiene rubber binders in graphitic and Si-based Li-ion electrodes with silver and bromine, enabling detailed electron imaging and precise spectroscopic quantification of the binder domain. Leveraging these methods, we perform binder-informed optimisation of electrode manufacturing, achieving a 14% reduction in electronic resistivity, suppression of binder migration during high-temperature electrode drying, and a 40% decrease in electrode ionic resistance. Furthermore, staining enables electrode-scale, high-resolution backscattered electron imaging of complex binder hierarchies, revealing multiple types of agglomerates and elusive nanoscale binder films. These films completely coat graphitic surfaces in pristine electrodes but shatter into highly inhomogeneous fragments after calendering in both research-grade and commercial electrodes, presenting new perspectives on interpreting common cycling stability and electrode performance issues. We show how binder staining can advance fundamental understanding, quality control and manufacturing optimisation of Li-ion electrodes, particularly those based on widely used water-processable binders.

Polymeric binders are critical in providing mechanical stability to all Li-ion battery electrodes, but occupy less than 5% of the electrode volume[1]. It is well known that the spatial distribution of the binder within an electrode strongly influences the cohesion and adhesion of electrode coating, cycling stability[2], electronic, ionic and thermal conductivity[3–6] and, thus, the longevity, rate performance and cost of Li-ion batteries[1,7]. Particularly for Li-ion battery negative electrodes, however, mapping the small binder fractions remains highly challenging, impeding rational, binder-informed optimisation of electrode structure, performance and manufacturing. During electrode fabrication, for example, drying-induced binder migration can severely undermine subsequent electrode stability and rate performance[8,9]. To prevent this, industrial roll-to-roll manufacturing of Li-ion electrodes uses relatively slow drying in 30–100 m-long ovens that dominate the footprint of a coating line[1,10,11] and require large capital investment (~105k euro/m of drying line, based on linear fitting of the published data[10]). As opposed to trial-and-error, efficient development of new drying approaches relies on accurate binder mapping techniques, as demonstrated in a number of studies[8,9,12–15]. Furthermore, among the many, often elaborate, techniques for decreasing tortuosity and increasing the power density of battery electrodes[16–19], changing binder distribution at the electrode scale presents an attractive[9], yet

[1]Department of Materials, University of Oxford, Oxford, UK. [2]The Faraday Institution, Didcot, UK. [3]Oxford Materials Characterisation Service, Department of Materials, University of Oxford, Oxford, UK. ✉e-mail: stanislaw.zankowski@materials.ox.ac.uk; patrick.grant@materials.ox.ac.uk

mostly unexplored alternative – because of the complex, hard-to-characterise nature of the binder domain. At a finer scale, the micro-distribution and coverage of binders on the active particle surfaces similarly remain challenging to determine[6,20–24], leaving a significant gap in understanding the effects of binders on key surface processes, such as Li intercalation[25] or formation of solid electrolyte interphase (SEI)[26–29].

Since the early 2000's, the water-soluble sodium salt of carboxymethyl cellulose (Na-CMC) blended with emulsified styrene butadiene rubber (SBR) has become the now-standard binder mixture used in industrial graphite and graphite/silicon negative electrodes[30], and is also being explored for next-generation Si negative electrodes[31] and water-processable LiFePO$_4$ positive electrodes[32,33]. Being dispersible in water rather than toxic N-methyl pyrrolidone (NMP), these binders considerably lowered the cost and complexity of negative electrode manufacturing, and improved cycling stability of these electrodes[1,34]. Contrary to the previous-generation polyvinylidene fluoride (PVDF) binder, however, SBR and CMC lack distinct, covalently bound elements that are traceable by readily accessible methods such as energy-dispersive X-ray spectroscopy (EDX) or backscattered electron imaging (BEI). Lacking a well-defined morphology, binders are also difficult to distinguish with scanning electron microscopy (SEM). Thus, there are only a few convincing demonstrations of mapping SBR and CMC, using, for example, thermogravimetry[15], Raman spectroscopy[35] (both requiring unpractically thick or concentrated samples) or more sophisticated time-of-flight secondary-ion mass spectroscopy[36]. Most recently, Lee et al. described staining of SBR with osmium tetroxide (OsO$_4$) that could be detected by energy-selective backscattered electron imaging (EsB) or EDX spectroscopy to trace SBR migration in graphite electrodes during drying[37,38]. The extreme toxicity and volatility of OsO$_4$, however, limit this approach to only the most specialised laboratories. Consequently, most of the works describing correlations between manufacturing conditions, electrochemical performance and binder distribution in graphite electrodes utilised EDX-traceable PVDF as a binder[8,9,12–14], which has different mechanical properties, binding mechanism and can show exaggerated migration during drying compared with the commercially-used CMC/SBR binders[15].

To circumvent these limits, here we describe two accessible binder staining methods that can be paired with EDX, BEI or EsB for rapid mapping and high-resolution imaging of aqueous binders in Li-ion battery electrodes. We demonstrate how these methods provide fundamental insights into binder organisation within the electrodes and enable binder-informed optimisation of Li-ion battery negative electrodes.

## Results

### Mechanism and specificity of staining reactions

Our concept of binder staining relies on exposing electrode samples to chemical agents that attach to the functional groups present within the binders but not in the other electrode components, as shown schematically in Fig. 1. Immersing graphite electrodes in aqueous AgNO$_3$ solution leads to Ag$^+$ ions binding to the carboxyl groups in CMC, forming a water-insoluble complex depicted in Fig. 1a. On the other hand, exposing the electrodes to Br$_2$ vapour brominates the aliphatic sp$^2$ carbons of SBR, as depicted in Fig. 1b. The latter reaction is similar to OsO$_4$ staining but with significantly lower hazard, as indicated by a 217-times higher short-term safe exposure limit for Br$_2$ compared with OsO$_4$[39]. Thus, the reactions can be performed using simple equipment − a Schlenk line for bromination (Fig. 1b and Supplementary Fig. 1), and a glass beaker and a vacuum desiccator for silverization (Fig. 1a). The mechanism of both reactions was confirmed using attenuated-total reflection infra-red spectroscopy (ATR-IR), X-ray photoemission spectroscopy (XPS) and EDX, showing diminished aliphatic C=C peaks in brominated SBR, a strong shift of -COO$^-$ peaks in silverized CMC and a substitution of Ag for Na, as discussed

in Supplementary Notes 1–2 and Supplementary Figs. 2–4. The high affinity of Ag$^+$ for carboxylate groups is not limited to CMC − silverization was also successfully applied to carboxyl-rich sodium alginate and sodium polyacrylate − two other widely researched aquoeus binders with a superior performance in e.g. silicon negative electrodes[31,34,40] (Supplementary Fig. 3). Note that because of the reactivity of Ag$^+_{(aq)}$ and Br$_2$, staining and binder mapping should be applied on separate electrode areas to those used in electrochemical or other testing.

To test the specificity of the staining reactions, we performed them on individual electrode components, i.e. dried films of CMC and SBR and powders of graphite and carbon conductive additive (C45). EDX and XPS were used to quantify Br and Ag concentration in the first 1 μm (EDX) and 10 nm (XPS) of the stained materials, showing a large amount of Ag in Ag-CMC and of Br in Br-SBR, and little to no binding of these elements to the other materials, as presented in Supplementary Fig. 3. Encouraged by these results, we used EDX to test the selectivity of staining on five electrode coatings containing graphite, 1 wt% C45 and 4% wt. (CMC + SBR) binder mixture with the relative CMC:SBR mass ratio ranging from 4:0 to 0:4. Each electrode coating was delaminated from the current collector, stained, and homogenised by pulverising in a mortar to remove any influence of binder migration on the quantification of staining efficiency. Figure 1c, d shows BEI of silverized and brominated electrode coatings with either only 4% CMC or 4% SBR, together with Ag and Br content quantified with EDX in all coatings in Fig. 1e. In the atomic number-sensitive BEI, the presence of Ag and Br strongly brightened contrast to the remaining materials containing mainly C, O and H. This allows clear identification of threads of Ag-CMC in the CMC-only electrode and patches of Br-SBR in the SBR-only electrode in Fig. 1c, d, respectively. Figure 1e shows that the amount of Ag in the silverized electrodes scaled linearly with the CMC content across all the CMC:SBR ratios. High selectivity of Ag to CMC over SBR was confirmed by the 16-times more Ag in the 4% CMC electrode compared to the minute Ag quantity in the 4 % SBR electrode. XPS analysis combined with detailed EDX mapping of non-homogenised electrodes presented in Supplementary Notes 2–3 and Supplementary Figs. 3–5 revealed that the small apparent reactivity of Ag to SBR was due to the reaction of Ag$^+$ with leftover reagents used in the manufacturer's SBR emulsion-polymerisation, such as ionic surfactants, persulfates and thiols[41,42], whose presence was indicated by Na, K and S found in the top of both SBR-only electrode and pure SBR sample. This analysis also showed that in non-homogenised electrodes, the negative impact of these reagents on the Ag-CMC staining selectivity may be larger in the top electrode region, where these compounds appeared to concentrate.

The amount of Br in the brominated samples also scaled linearly with the SBR content in the electrodes, with 7-times more Br in the 4% SBR electrode compared to the 4% CMC electrode (Fig. 1e). The small amount of Br binding to CMC was due to surface oxidation of Na-CMC by Br$_2$ and binding of Br to Na (see exemplary Hunsdiecker reaction scheme in Fig. 1b)[43]. The associated formation of NaBr was suggested by the 1:1 elemental ratio of Br to Na detected by local EDX mapping inside the porous, high surface-area CMC/C45 agglomerates, as well as the mostly ionic (Br$^-$) form of bromine detected with XPS in the top few nanometres of brominated CMC, as presented in Supplementary Notes 2,4 and Supplementary Figs. 3–4 and 6. The formation of NaBr on the CMC surface enabled BEI observation of CMC in brominated electrodes (Fig. 1d), similarly to Ag binding to CMC-COO$^-$ groups in silverized electrodes (Fig. 1c). The ionic binding of Br in brominated CMC was also stable under the SEM/EDX electron beam, contrary to the less stable Br bound to polymeric SBR, which progressively outgassed from the SBR-containing electrodes during continuous electron irradiation, as visible in time-resolved Br tracing in Fig. 1f. The extent of Br outgassing scaled linearly with SBR amount in the samples (Fig. 1g), suggesting it could be used to verify SBR

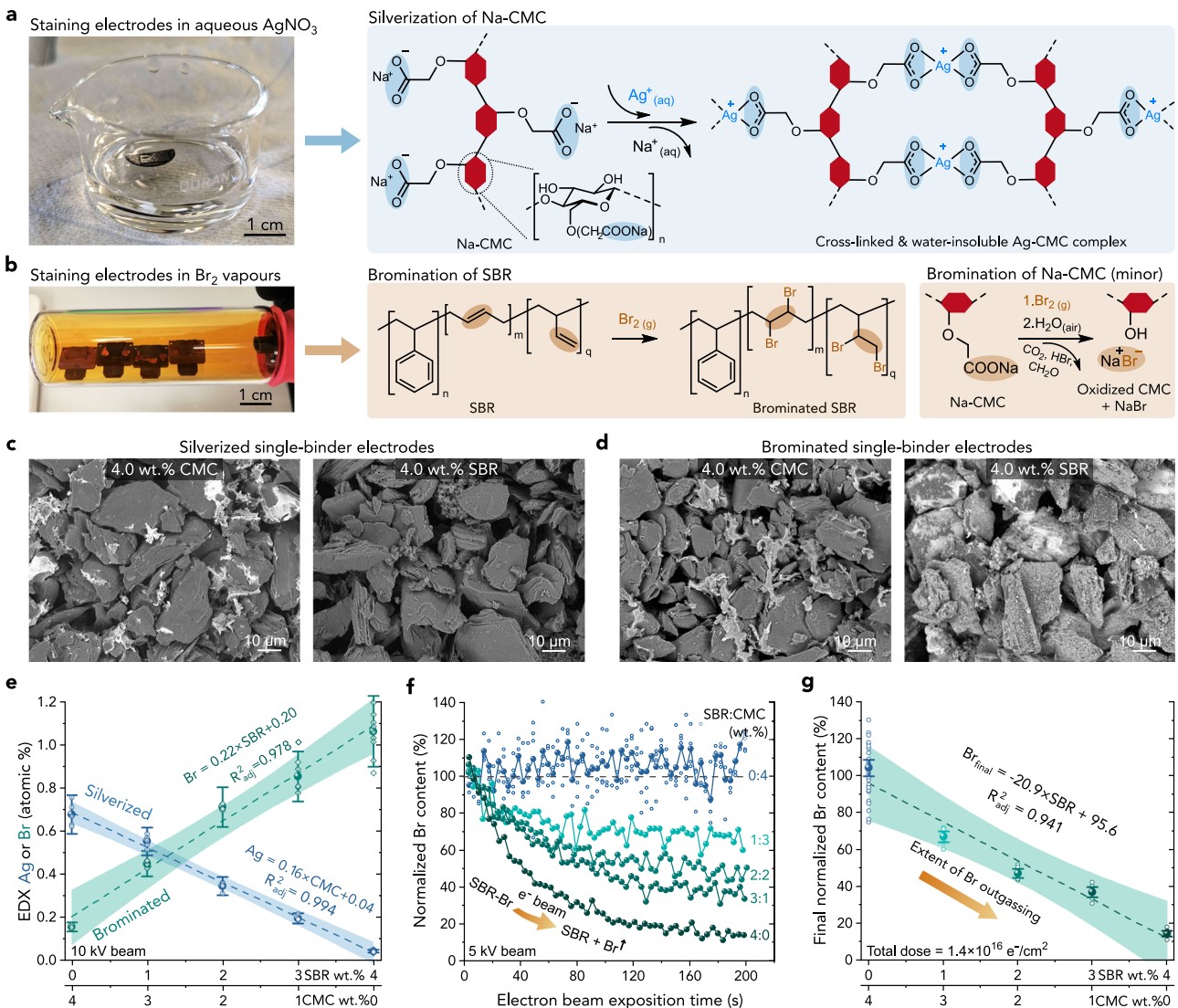

**Fig. 1 | Mechanism and selectivity of the staining methods. a** Photograph of a supported fragment of a graphite electrode immersed in aqueous $AgNO_3$ during silverization, together with a reaction scheme of $Ag^+$ binding to Na-CMC. **b** Photograph of supported electrode fragments during bromination, together with a reaction scheme of Br binding to SBR (primary reaction) and Na-CMC (side reaction). **c, d** Contrast-equalised BEI of Ag- and Br-stained and homogenised graphitic electrodes containing either 4 wt% CMC or 4 wt% SBR. **e** Ag and Br atomic % determined with EDX in stained and homogenised electrodes containing 4 wt% of CMC + SBR mixture with different CMC:SBR ratios. The number of technical replicates (1.46 × 1.09 mm area measurements) was $n = 5$ for all the samples except for Br in 3% SBR ($n = 7$) and in 4% SBR ($n = 9$, excluding one outlier flagged in the Source Data). **f** EDX Br content in brominated electrodes during continuous irradiation

with SEM/EDX electrons, normalised to the average Br content in the first 10 and 35 s of exposure for the 1–4% SBR and 0% SBR coatings, respectively. The number of technical replicates (152 × 114 µm area measurements) was $n = 1$ in all electrodes except for 0% SBR, where $n = 4$ for better signal-to-noise (individual and averaged measurements of this sample are shown as small and large blue points, respectively). **g** Normalised Br content at the end of measurements in **f**, averaged between 170 and 200 s ($n = 34, 9, 9, 9, 8$ for the 0-4% SBR samples, respectively). In **e, g**, individual measurements and means are shown with small and large markers, respectively, and the error bars and shaded areas represent 95% confidence intervals of means and linear fits, respectively. The samples in **c**−**e** were silverized with the addition of a surfactant, as explained in "Methods". Source data are provided as a Source Data file.

quantification in brominated electrodes containing CMC − as we demonstrate below.

## Staining bi-layered electrodes

Having established the high specificity of the staining reactions, we tested their sensitivity to varying binder fraction within graphite electrodes, as presented in Fig. 2. This was verified using a model, -300 µm-thick bi-layer electrode spray coated onto a Cu foil using a method reported in our previous work refs. 44–46. By consecutively spraying and instantly drying two slurries with relatively high and low binder concentration, we designed this electrode to have a 4-times difference in binder fraction between a lower, binder-rich layer (with 4.5% SBR and 3.0% CMC by weight) and an upper, binder-lean layer

(with 1.13% SBR and 0.75% CMC), keeping the overall binder content within the typical range of Li-ion electrodes. After fabrication and calendering, electrode fragments were stained with either $Br_2$ or $Ag^+$, cross-sectioned using Ar plasma milling, and analysed using EDX and BEI. Note that before silverization, the Ag-treated electrodes were detached from the current collector to avoid reaction of $Ag^+$ with the Cu foil current collector, as explained in "Methods".

The EDX maps of Ag-L and Br-L X-ray series presented in Fig. 2a show an expected bi-layer distribution of the staining elements, which is reflected in the derived through-thickness Ag and Br profiles in Fig. 2b. The profiles were individually ratioed and indicate 3.9 ± 1.8 (silverized) and 3.1 ± 0.4 (brominated) average element ratio $\Delta$ between the binder-rich and binder-lean layers, also summarised in

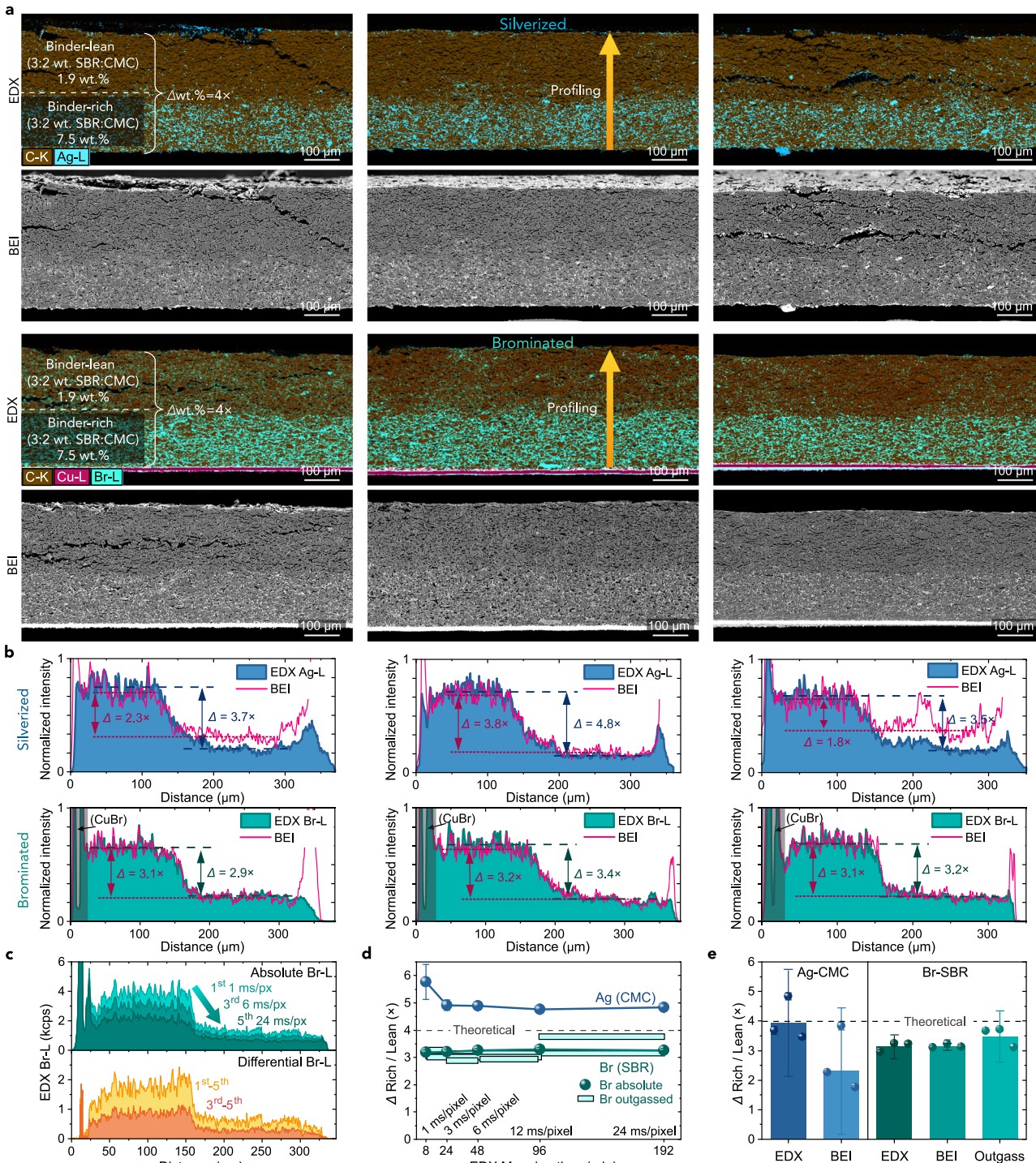

**Fig. 2 | Sensitivity of binder mapping in bi-layered electrodes. a** EDX maps and contrast-equalised BEI images of silverized and brominated electrodes with bi-layer binder distribution. **b** EDX- and BEI-derived through-thickness profiles of Ag and Br, showing ratios ($\Delta$) of each signal between the binder-rich and binder-lean layer. **c** EDX-Br profiles of the third brominated sample acquired consecutively at an increasing pixel scanning time (total electron dose), also showing Br outgassing profiles as differentials between given scans. **d** Accuracy of $\Delta$'s vs EDX mapping time (pixel dwell time) in the second silverized and third brominated sample, derived from the absolute Ag and Br signals (points) or outgassed Br between different scan pairs (horizontal bars). Each point and horizontal bar represents a single ($n = 1$) $\Delta$

from one scan or a pair of scans, respectively. The $\Delta$'s were calculated from mean signals in 110 µm (120 data points) of the binder-rich and lean layers. **e** $\Delta$'s derived using EDX, BEI and Br outgassing in all samples. For each staining type, $n = 3$ electrode samples were individually stained and measured with EDX and BEI. Weighted means (columns) were averaged from the individual measurements (points). The error bars correspond to Student's $t$-95% confidence intervals of derived $\Delta$'s (**d**) or averages of $\Delta$'s (**e**). All images were acquired at 10 kV and post-processed as described in "Methods". Source data are provided as a Source Data file.

Fig. 2e (note, ±corresponds to 95% confidence intervals). These average values are within 2.5% and 22% of the expected 4× ratio, with more undercounted $\Delta$ for the Br-stained SBR due to the larger susceptibility of SBR to drying-induced migration amplified by its weaker adsorption onto graphite[47], as verified in Supplementary Note 5 and Supplementary Fig. 7. The element ratios in individual samples were also within 5–23% (silverized) and 4–6% (brominated) of the average values, indicating good precision in single 1 mm-wide cross-section measurements. The addition of a surfactant (1 mM Triton X-100) to the AgNO₃ staining solution benefited wetting and accuracy of Ag-CMC profiling through the binder-lean region of the samples, as discussed in the wetting analysis in Supplementary Note 6 and Supplementary Fig. 8. The higher complexity of liquid-phase silverization and lower intrinsic Ag signal in Ag-CMC (as quantified in Fig. 1e) could also explain the larger spread of silverized $\Delta$'s compared to the brominated ones (Fig. 2e).

For the Br-SBR analysis, we verified its accuracy by following Br outgassing. This was done by subtracting Br-L traces derived from EDX maps acquired consecutively at increasing pixel dwell time (corresponding to the total mapping time and total electron dose). Fig. 2c shows that the differential (outgassed) Br-L profiles closely followed the absolute Br-L profiles, with a similar outgassing $\Delta$ of $3.5 \pm 0.9\times$ averaged across the three samples (Fig. 2e). Also, for the absolute EDX Ag-L and Br-L profiling, increasing mapping time did not significantly change the $\Delta$ values - accurate through-thickness binder profiling across these ~350 μm-thick electrodes could be performed as rapidly as 8 min for Br and 24 min for Ag (Fig. 2d). Although generation of higher quality EDX maps required at least 1 h (e.g. 3 h for the maps in Fig. 2a), the mapping time could be reduced to just a minute using BEI (Fig. 2a), due to much higher yield of backscattered electrons than electron-generated X-rays[48]. By analysing the distribution of white pixels in BEI images after multilevel Otsu thresholding[49], they could also be used for quantitative binder profiling. Except for two silverized samples, the BEI-derived pixel profiles in Fig. 2b matched the EDX elemental profiles and gave similar $\Delta$ values. However, cross-sectional BEI remained more uncertain due to its sensitivity to image artefacts (e.g. particle surface charging) and trace metallic contamination after ion polishing, as shown in Supplementary Figs. 9 and 10.

## EsB imaging of local binder morphology

Having established the staining methodology, we investigated detailed local binder organisation inside graphitic electrodes by combining staining with EsB. EsB is a low beam energy-variant of BEI that utilises a polarised filtering grid in front of an in-lens backscattered electron detector to capture backscattered electrons originating only from a shallow depth of the analysed surface, with high compositional resolution[50–52]. The method has been used previously for mapping the general distribution of PVDF and OsO₄-stained SBR in graphite electrodes, but without focusing on local binder morphology[37,52].

Figure 3 shows secondary electron and EsB images of the interior of an uncalendered standard graphite electrode (with 2.2 wt% SBR and 1.5 wt% CMC) after silverization. The images were acquired at low and high magnification (Fig. 3a, b, respectively) and gradually decreasing beam voltages, corresponding to decreasing electron-sample penetration depth and increasing surface sensitivity in consecutive images[48]. At 10 kV, the EsB images show bright, few micron-sized agglomerates of stained CMC and C45 conductive carbon nanoparticles, localised mostly between graphite particles (denoted as $\alpha$ phase in Fig. 3a, b). Containing a large number of C45 nanoparticles, these agglomerates influence long-range interparticle electronic conductivity within the electrode[53] (also see the "Correlating slurry mixing, CBD agglomeration and the electrical resistivity of graphite electrodes" section below). The contrast from these agglomerates at 10 kV originated from a relatively bulky stained binder within their composite structure. Consequently, this contrast mostly disappeared at lower

beam voltages of 2.5 and 2 kV (shown in Fig. 3a), where the depth probed by backscattered electrons was reduced from a few hundred to a few dozen nanometres (see the simulations in Fig. 3c). At these lower beam voltages, a second agglomerate type containing tightly packed, dark SBR nanoparticles mixed with CMC and C45 became visible ($\beta$ morphology in Fig. 3a, b). The differentiation between C45, CMC and SBR was achieved by comparative analysis of electrodes prepared without SBR or C45, presented in Supplementary Fig. 11. Due to the large concentration of SBR in these agglomerates, they provide mechanical cohesion and elasticity to the electrode[54–56]. Finally, the EsB images acquired at even lower beam biases of 1.5 kV and 1 kV surprisingly revealed a third, continuous-layer morphology of binder (termed $\gamma$ in Fig. 3a, b) that coated large regions of graphitic surfaces and was not visible at larger beam voltages. This binder layer was also not visible in control EsB images of a non-stained electrode or of a silverized CMC-free electrode, being only visible in the CMC-containing electrodes (Supplementary Fig. 12). These comparator experiments confirmed that bright EsB contrast at 1.5 and 1 kV in Fig. 3 unambiguously corresponded to a continous thin film of silverized CMC that also embedded some non-stained (dark) SBR nanoparticles, visible in Fig. 3b. These observations were reproduced in several hundreds of EsB images acquired on four separately-made graphite electrodes after staining, some of them shown in Supplementary Fig. 13.

Monte Carlo simulations were used to estimate the thickness of the CMC layer based on its changing visibility at different EsB imaging voltages, as described in "Methods". Figure 3c shows the simulated depth of electron backscattering events in an Ag-CMC layer on a graphite particle at different electron beam energies. As an approximate detectability criterion, we assumed that the proportion of the electrons backscattered in the Ag-CMC layer to the electrons backscattered in the underlying graphite should be much larger than 0.5 at 1 keV and 1.5 keV for detection, and similar or lower than 0.5 at 2 keV and 2.5 keV for no detection. The plots of the cumulative distribution of backscattered electrons in Fig. 3c show that this condition was met for an Ag-CMC thickness of only 10-15 nm, which is in the same range as the thickness derived from high resolution SEM/EsB imaging of a fractured graphitic particle (Supplementary Fig. 14), as well as the assumed uniform binder layer thickness in graphitic electrodes calculated by Landesfeind et al.[57]. Because the CMC layer was so thin, it closely followed the morphology and contours of the underlaying graphite surfaces. This is why the layer could not be distinguished from bare graphite using conventional secondary electron microscopy (both in this and in previous works), even at very low beam voltages (Fig. 3b). Also note that in the ion-polished samples, the CMC layer was completely masked by trace metallic contaminants deposited on graphitic surfaces after the polishing (Supplementary Fig. 10). The thin layer could also be observed in EsB images of silverized electrodes with an analogous polyacrylate binder (Supplementary Fig. 15). Previous works have postulated that binders form nanoscale layers on active particle surfaces in Li-ion electrodes[6,58–61], but the visual evidence for such binder morphology, until now, was limited to only a few higly localized spectroscopic transmission electron microscopy (TEM) studies[22–24].

## Binder coverage on active particle surfaces

The EsB images in Fig. 3a, b were acquired from an internal, delaminated region of an electrode where the bright CMC layer appeared as a patchy layer, allowing for differentiation from dark uncoated graphite surfaces. To assess if delamination might have affected the binder distribution, we also acquired images from the electrode top (not delaminated) surface (Fig. 4). The images of the silverized electrode in Fig. 4a show that the CMC layer completely coated graphite particle surfaces, with additional brighter patches of Ag bound to CMC or SBR emulsifiers that migrated to the surface during electrode drying. These EsB images also show additional dark areas

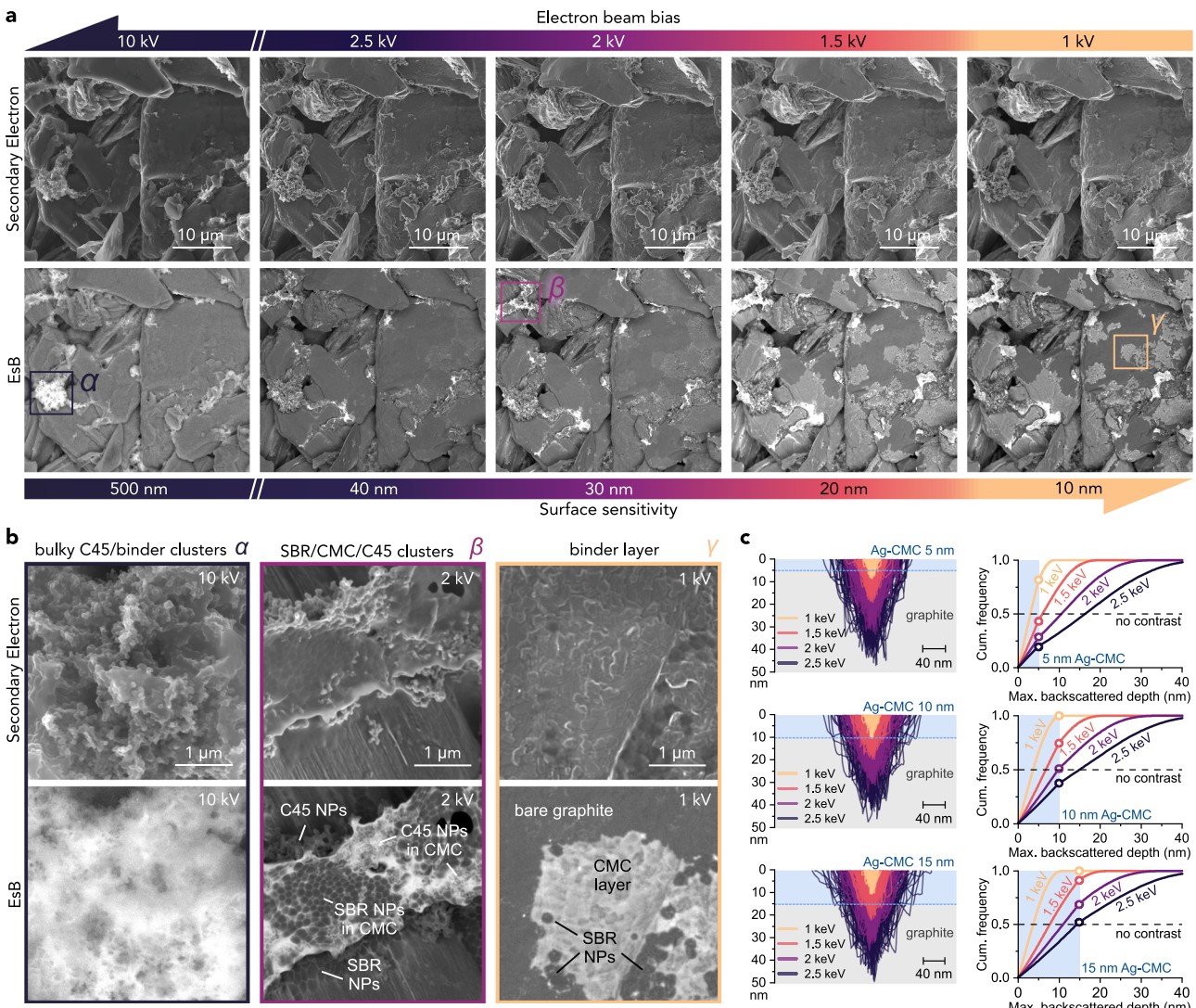

**Fig. 3 | Secondary electron and EsB imaging of local binder morphology inside silverized graphite electrodes. a** Images of internal electrode surface exposed by scotch tape peel-off and subsequent silverization, imaged at different electron beam voltages. **b** High-magnification images of the three types of binder morphology (termed α, β, γ) detected in **a**. **c** Monte Carlo-simulated trajectories and depth distributions of backscattered electrons in layered Ag-CMC/graphite at different primary electron energies and Ag-CMC thicknesses. All the EsB images were acquired at an EsB filter grid voltage of −513 V. Therefore, the simulations in **c** only consider backscattered electrons with an exit energy larger than 513 eV. The electrodes contained 2.2 wt% SBR and 1.5 wt% CMC. Simulations were performed using the Casino package[81]. The description of electrode preparation and simulation procedure is provided in "Methods". Source data are provided as a Source Data file.

that were not stained by $Ag^+$ but produced charging in the secondary electron images (see the marked areas in Fig. 4a). As shown in Fig. 4b, on bromination, these areas appeared bright in the intially-acquired EsB images and rapidly faded in subsequent images. Recalling the electron-induced outgassing of Br from Br-SBR, this allowed indentifying these patches as SBR. Additionally, high magnification images show that they consisted of self-assembled individual SBR nanoparticles (the same, less agglomerated SBR particles were also visible in the CMC layer in Fig. 3b). That SBR retains nanoparticle form in the electrode results from the mild mixing conditions required during slurry preparation to prevent SBR demulsification[62]. The island-like agglomeration of SBR compared to the uniform layer appearance of CMC also evidences their different affinity to graphite surfaces[47]. Furthermore, Br outgassing from the "free" (not agglomerated with CMC) brominated SBR nanoparticles differentiates them from the superficially brominated CMC layer, whose EsB contrast remained unchanged during consecutive image acquisitions (see the images of an internal part of the brominated electrode in Supplementary Fig. 16). The outgassing of Br-SBR also allowed for quantification of

SBR particle coverage by subtracting a later-acquired EsB image from the first EsB image (see Methods). We estimate that the ~120 nm thick SBR agglomerates covered 8–9% of the apparent graphite surface area on top and within this electrode, with an additional 6% of the internal electrode surfaces covered by the SBR/CMC agglomerates. The similarity of binder morphologies detected with EsB inside silverized (Fig. 3) and brominated (Supplementary Fig. 16) electrodes also suggested that the binder distribution itself was not affected by the staining procedures.

During the manufacturing of Li-ion electrodes, slurry coating and drying are followed by compressing (calendering) the electrodes between two metal rolls. Calendering reduces inter-particle porosity and often improves electrode electrical conductivity[63], but its effect on binder distribution is largely unknown. Thus, we analysed binder surface coverage after hot calendering the electrode to 30% porosity at 80 °C. The EsB images from the top and interior of the calendered electrode in Fig. 4c show fracture and delamination of both the γ binder film and β SBR/CMC/C45 agglomerates. Fracture resulted in large, irregularly distributed areas of graphite particles where the

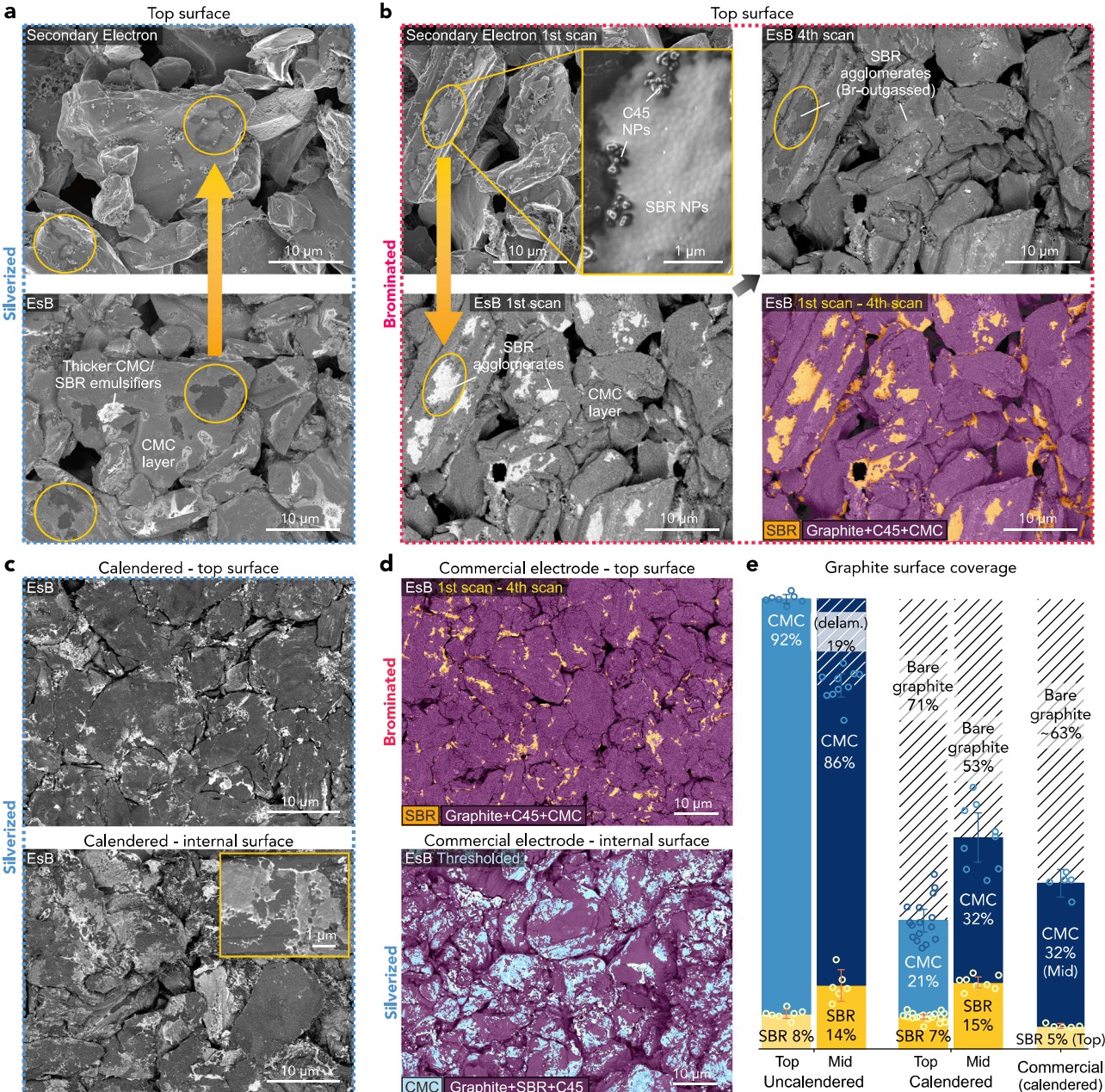

**Fig. 4 | EsB imaging of local binder morphology at the top and within the graphitic electrodes before and after calendering. a** Top silverized uncalendared electrode surface. **b** Top brominated uncalendared electrode surface imaged in the first and fourth EsB scan. The colour-enhanced image shows the results of image segmentation based on the difference between the EsB images acquired in the first and fourth scans. **c** Top and internal electrode surface after calendering and silverization, showing cracking and delamination of the CMC layer. **d** Colour-enhanced EsB images of the top and internal surfaces of a commercial electrode after bromination and silverization. **e** Mean binder coverage of graphite surfaces on top and inside electrodes before and after calendering, determined through contrast filtering. Each binder was stained on separate pieces of uncalendared, calendered and commercial electrodes and its coverage was determined through contrast filtering of *n* EsB images (technical replicates), except for the CMC in the Uncal. Top electrode, whose coverage was calculated as 100%-SBR%. The values of *n* were, for SBR: 7, 6, 18, 8, 5 in the Uncal. Top and Mid, Cal. Top and Mid, Commercial Top, respectively; for CMC: 7, 10, 16, 9, 5 in the Uncal. Top and Mid, Cal. Top and Mid, Commercial Mid, respectively. The 86% CMC coverage in the Uncal. Mid electrode includes the 19% of bare graphite surfaces from scotch-tape delamination. The error bars represent 95% confidence intervals of the mean values. All images were acquired at 1 kV of beam voltage and post-processed as described in "Methods". Source data are provided as a Source Data file.

previously near-complete CMC layer was shattered and partially delaminated. The remaining CMC patches covered only 32 ± 5% of the particle surface, based on gaussian filtering and grey-level thresholding of five EsB images (Fig. 4e). On the top electrode surface directly exposed to the hot calender rollers, CMC coverage was even lower (21 ± 3%, also visible in Fig. 4c). The shattering of the CMC layer was consistent with the reported brittleness of dried CMC[64]. These findings are not limited to laboratory-made electrodes—we

also imaged uncycled commercial electrodes from LiFUN Technology (Xinma Industry Zone, Hunan Province, China). Figure 4d shows identical shattered CMC layers and SBR agglomerates in these electrodes, with over 60% of graphitic surfaces left bare and uncoated (Fig. 4e). Such high inhomogeneity in binder surface coverage can be expected to strongly affect surface processes on active material particles, such as Li⁺ (de)intercalation or formation of SEI, as explained in detail later.

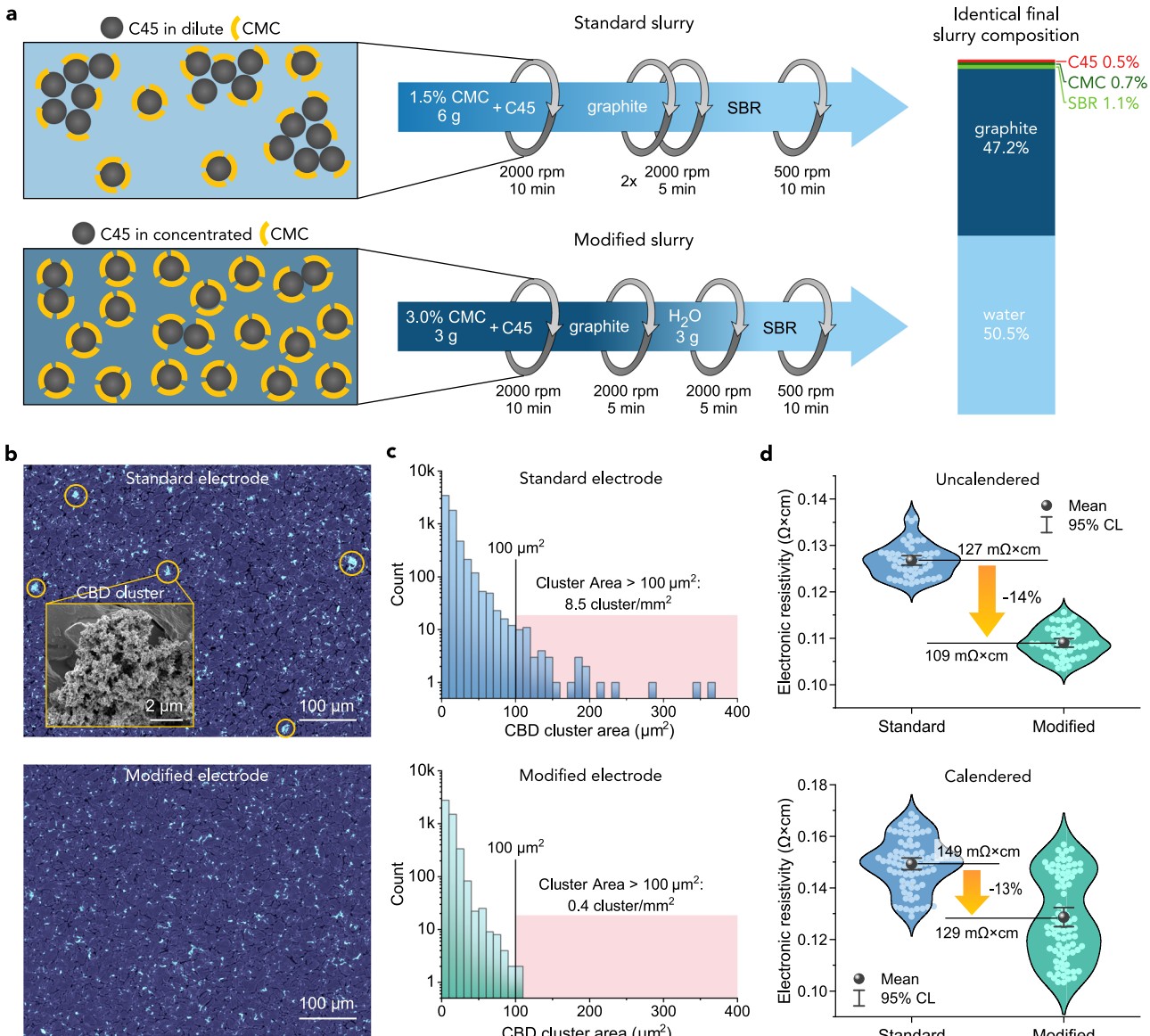

**Fig. 5 | Correlation between slurry processing, carbon-binder domain (CBD) agglomeration and electrical conductivity of the electrodes. a** Schematics of the preparation and mass composition of electrode slurries using initially dilute 1.5% wt. CMC (standard slurry) or initially concentrated 3% wt. CMC (modified slurry). In both protocols, each 2000 rpm mixing step was preceded by 1 min premixing at 500 rpm; mixing was done in a Thinky planetary centrifugal mixer. **b** Colour-enhanced top-surface BEI of brominated electrodes prepared from standard and modified slurries. The inset shows an SEM image of a CBD cluster. **c** Logarithmic histograms of CBD cluster size distribution in standard and modified electrodes, based on analysis of 10 BEI images of each electrode. **d** Violin plots showing the distribution of electronic resistivity of standard and modified electrode coatings before and after calendering, determined by four-point probe measurements. Five (uncalendared) and eight (calendared) samples of each coating were measured ten times each. Sample properties are summarised in Supplementary Table 1. BEI image segmentation and CBD clustering analysis were done as described in "Methods". Source data are provided as a Source Data file.

## Correlating slurry mixing, CBD agglomeration and the electrical resistivity of graphite electrodes

To demonstrate how binder mapping can benefit the development of Li-ion electrodes, we carried out two practical studies. First, we investigated the relationship between the conditions of slurry processing, the distribution of conductive carbon/binder domain (CBD) within the electrodes, and electrode electrical resistivity (Fig. 5). In the graphite electrodes prepared using our standard slurry mixing protocol shown in Fig. 5a, large CBD clusters could be readily detected after bromination or silverization with BEI (Fig. 5b) and EsB at 10 kV (Fig. 4a). Although the spatial density of these clusters was moderate (8.5 cluster per mm²), they had a large geometric area of more than 100 μm²/cluster, shown in the associated histogram in Fig. 5c. This implied that a fraction of binder-embedded C45 conductive carbon

particles was ineffectively and inhomogeneously distributed across the electrode. To reduce CBD agglomeration, we fabricated electrodes of the same dry mass composition, but using a more concentrated CMC solution (increased from 1.5 to 3 wt%) during an initial mixing step. After subsequent mixing with graphite, the slurry was diluted with water to the same final water fraction as in the standard slurry, as shown in Fig. 5a. We hypothesized that because the amphiphilic CMC electrostatically stabilizes solids dispersion in the slurry by adsorbing onto the graphite and carbon particle surfaces[47], a more concentrated CMC solution in the first mixing step might increase CMC saturation on C45 surfaces, improving C45 dispersion in the slurry and in the final electrode. Figure 5b shows a BEI image of such a modified electrode after bromination, confirming the disappearance of large CBD clusters, which is also evident in the histogram in Fig. 5c. Interestingly, such a

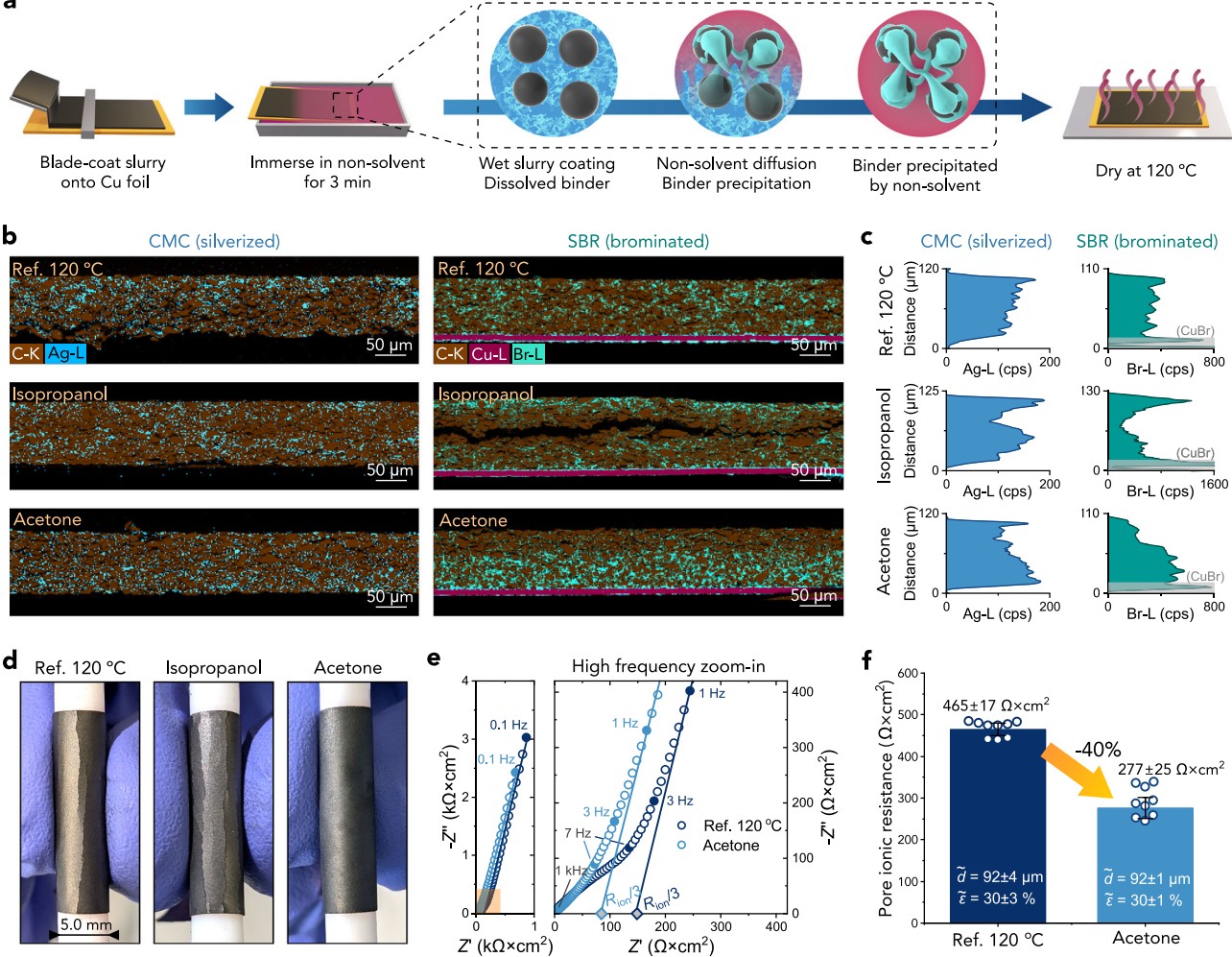

**Fig. 6 | Phase inversion effects on binder distribution, mechanical cracking and pore ionic resistance of graphitic electrodes. a** Schematic of the phase inversion process. **b** EDX cross-sectional mapping of Ag-stained CMC and Br-stained SBR in a reference electrode dried at 120 °C and in electrodes phase inverted in isopropanol and acetone prior to drying at 120 °C. **c** Through-thickness distribution of Ag-stained CMC and Br-stained SBR in the electrodes, integrated and smoothed from EDX maps over 1 mm-wide cross sections. The Ag-L (CMC) profile of the reference electrode was corrected (flattened) by subtracting a positive gradient detected in the reference C-K signal of this sample, as described in "Methods". **d** Photographs showing cracking of the reference and isopropanol-processed electrodes and no damage in the acetone-processed electrode on bending around a 5 mm-thick rod. **e** Electrochemical impedance spectroscopy (EIS) of reference and acetone-processed electrodes measured in symmetric cells. The EIS spectra were acquired

in 10 mM tetrabutyl ammonium perchlorate in 1:1 wt. EC:DMC, between 100 kHz and 10 mHz; the series resistance-subtracted impedance was halved to reflect a single electrode. The lines represent linear fitting of the data between 3 Hz and 0.1 Hz used for the derivation of pore ionic resistance ($R_{ion}$). **f** Mean pore ionic resistance of the reference and acetone-processed electrodes. For both the reference and acetone-processed electrode, 3 symmetric cells (each made of two independently-calendered 2.01 cm²-sized electrode fragments cut from a 100 cm² parent electrode) were measured 3 times each ($n = 9$), and the results were averaged. The average thickness ($d$) and porosity ($\varepsilon$) were identical for both types of electrodes and are indicated in **f**. The error bars and ±values correspond to 95% confidence intervals of the averages. Source raw and processed data are provided as a Source Data file.

level of CBD de-agglomeration could not be achieved using zirconia balls during mixing of the standard 1.5 wt. % CMC slurry, indicating a stronger influence of the initial CMC concentration on the CBD dispersion (Supplementary Fig. 17). Recently, Park et al. similarly found that increasing the initial CMC concentration during mixing with carbon black particles improved their dispersion in wet slurries[65]. Furthermore, four-point probe measurements of not-stained samples showed that reduced CBD agglomeration decreased mean electronic resistivity in both uncalendared and calendered electrode coatings by $14 \pm 2\%$ (Fig. 5d), albeit with a somewhat larger uncertainty after calendering. While the absolute electronic resistivity of both electrode coatings was low ( ~ 0.1 Ω × cm) due to the omnipresence of electronically conductive graphite, these findings could help to optimise other electrodes based on more resistive active materials, e.g. LiFePO₄ or sulfur, and similar binder/carbon additives. Future staining-assisted

optimisation could investigate alternative pre-mixing methods of thinner, lower CMC-content slurries, or substituting C45 nanoparticles with carbon nanotubes or graphene for further resistivity decrease[6].

## Mitigating binder migration during rapid electrode drying through phase inversion

In the second application of binder staining, we explored phase inversion as a method to prevent binder migration during fast, high-temperature electrode drying (Fig. 6). Phase inversion is a roll-to-roll compatible process, where a polymer solution (e.g. PVDF in NMP) is immersed in a non-solvent (e.g. water) that destabilises the solution and causes rapid precipitation of the polymer, as schematically shown in Fig. 6a[66]. Several studies demonstrated that this process can improve manufacturability and cycling performance of >100 μm thick PVDF- and poly(ether sulfone)-based battery electrodes, partly due to

induced directional porosity or altered PVDF morphology[67–69]. However, the effect of precipitating binders before electrode drying on final binder distribution has been difficult to assess. Therefore, we first screened 17 potential non-solvents and found that acetone and isopropanol rapidly precipitated CMC from aqueous solution, as shown in Supplementary Fig. 18. Then, we applied phase inversion to freshly-coated, wet graphite electrodes by dipping them for 3 min in isopropanol or acetone before drying at 120 °C and calendering at 80 °C, and compared them with a reference electrode processed identically but without phase inversion. Following staining of the binders, the EDX maps and binder profiles in Fig. 6b, c show that high-rate drying increased the CMC and SBR fraction at the top of the reference electrode. Blocking of the top surface pores with excessive binder obstructs the Li+ access to the electrode volume and leaves less binder at the interface with the current collector, lowering electrode adhesion[9]. Nonetheless, the migration was not as severe as in PVDF-based electrodes, in agreement with the recent results of Lee et al.[37]. Figure 6b shows that immersion in isopropanol further concentrated the binders at the top electrode surface by forming a dense polymer "skin layer", likely associated with a relatively slow isopropanol permeation and delayed precipitation of the binder[66,67]. On the other hand, the electrode immersed in acetone showed a marked binder concentration near the current collector, possibly due to faster precipitation of CMC by acetone with 6× lower viscosity than isopropanol[70]. The greater binder fraction at the current collector allowed to extensively bend this relatively thick electrode without cracking, making it significantly more adherent to the current collector compared with the two other electrodes with the opposite binder distribution, as tested on not-stained samples (Fig. 6d). The EDX results were reproducible on separately made, processed and stained electrodes (Supplementary Fig. 19).

To explore effects on pore ionic resistance, we tested unstained samples using electrochemical impedance spectroscopy (EIS) in a non-intercalating electrolyte developed by Landesfeind et al. [3,57]. Figure 6e shows the Nyquist plots of the reference and acetone-processed electrode in which the pore resistance was derived by extrapolating the low-frequency, near-linear regions of the plots[4]. Figure 6f shows that the measured ionic resistance was 40% lower for the acetone-processed electrode compared to the reference electrode, with the actual resistance difference possibly even larger[71], as discussed in "Methods". The process did not change the comparatively low electronic contact resistance of the coatings, indicated by the absence of a high-frequency semicircle in the Nyquist plots[3,72] (highlighted in Supplementary Fig. 20). Considering that all electrodes had identical composition, thickness and total porosity, the marked reduction in ionic resistance was a sole consequence of the preferential binder (and porosity) distribution, in agreement with simulations[9,73]. As demonstrated by Morasch et al. on a high-temperature-dried, mechanically delaminated and upturned electrode with a similar final binder distribution, reduced pore resistance leads to significant improvements in electrode fast charging and discharging performance[9], and likely extends electrode lifetime[74,75]. Our results show that the same preferential binder distribution and reduced ionic resistance could be realised with a much more practical and potentially up-scalable phase inversion, which simultaneously enables rapid electrode drying. Combining phase inversion with using spherical (instead of flaky) graphite or pore patterning techniques could further improve ionic transport in such electrodes[5,16–19].

### Staining limitations

Finally, we outline the present limitations of this study. First, the staining methods, particularly the liquid-phase silverization, are less applicable to electrodes containing easily-oxidizable materials (e.g. metallic, LiFePO$_4$ or nano-Si particles), as these can rapidly undergo redox reactions with the staining elements, complicating image interpretation. Micro-Si or materials with an inert, stable surface passivation layer can be, however, suitable. This is shown in Fig. 7, which demonstrates successful staining on Si and C-coated SiO$_x$ microparticle-based electrodes but not on the more reactive LiFePO$_4$ or nano-Si. For the same reason, silverization requires impractical detachment of any Cu current collector prior to staining, and the likely alteration of the electrode's electrochemistry by Ag+ and Br$_2$ staining may restrict its applications, for example, in operando studies or online manufacturing quality control. Also, despite the time- and accessibility advantages of BEI, this imaging mode has a limited applicability for mapping Ag/Br-stained electrodes containing other high-Z elements (e.g. Si) where the use of EDX may be necessary (see BEI vs EDX in Fig. 7c). The same limit applies to EsB, where the contrast remains largely relative, highly sensitive to surface contamination, and requires calibration (as demonstrated herein)[50]. Furthermore, to minimise geometrical artefacts, the cross-sectional binder analysis requires lengthy sample plasma cross-sectioning (~8–12 h for 100 µm-thick electrodes), which might be shortened using laser pre-cutting or resin impregnation and microtomy. Also, because Ag+/Br$_2$ staining targets carboxylate and aliphatic C=C groups in the binders, processes that alter these groups beforehand (e.g. drying-induced cross-linking) may reduce Ag/Br uptake (which, however, could be exploited to quantify binder cross-linking). Consequently, bromination and silverization are not suitable for staining e.g. PVDF or PTFE binders (that lack these functional groups), but can be suitable for other binders, such as the carboxyl-rich alginates and polyacrylates (as shown in Supplementary Fig. 15). However, the accuracy of spatial mapping of these two binders in electrodes remains to be fully quantified.

## Discussion

We have developed simple staining methods for mapping common aqueous binders – CMC and SBR – in graphitic Li-ion electrodes and showed how they enable better fundamental understanding and optimisation of these electrodes. On exposing electrodes to aqueous Ag+ or gaseous Br$_2$, these elements bind to CMC and SBR, respectively, and act as markers for binder mapping using EDX (longer, but more accurate) and BEI (rapid, but less accurate), as verified on model bi-layered electrodes. Because Br binds to both SBR and (superficially) to CMC, bromination could be used for simultaneous BEI imaging of both of the binders, and the outgassing of Br from SBR in consecutively-acquired images allows for its distinction from CMC. The methods are compatible with micro-Si and SiO$_x$-based electrodes, as well as analogous unsaturated and carboxylated binders, such as alginates and polyacrylates. We have shown that simple processing interventions and seemingly small differences in binder distribution can profoundly impact key metrics of the otherwise identical electrodes, improving their mechanical toughness or electronic and ionic conductivity. Staining combined with BEI, in particular, is simple and attractive for binder-informed optimisation and off-line quality control of electrodes in industrial settings, considering the availability of BEI even in table-top SEMs. The cost of implementing these methods comprises capital and operational costs of EDX/BEI-equipped SEM, while the staining itself utilises simple equipment and very small quantities of relatively cheap and reusable chemicals (here, 0.3 g AgNO$_3$ or 0.6 g Br$_2$ per batch of up to 5 samples). With optimisation of sample-cross sectioning, the complete staining, polishing and through-thickness EDX analysis time may be reduced from 15 to 5 h/sample and down to 3 h/sample for BEI analysis. Note that the staining and analysis, not requiring cross-sectioning, such as top-down CBD agglomeration check with BEI, can already be performed in less than 1 h/sample.

Combined with EsB, the staining methods enabled large-scale observation of surprisingly complex morphologies of CMC and SBR in laboratory-made and commercial graphitic Li-ion electrodes. The access to electrode-scale, high-resolution imaging of nanoparticulate SBR agglomerates and nano-thick CMC conformal films on graphitic

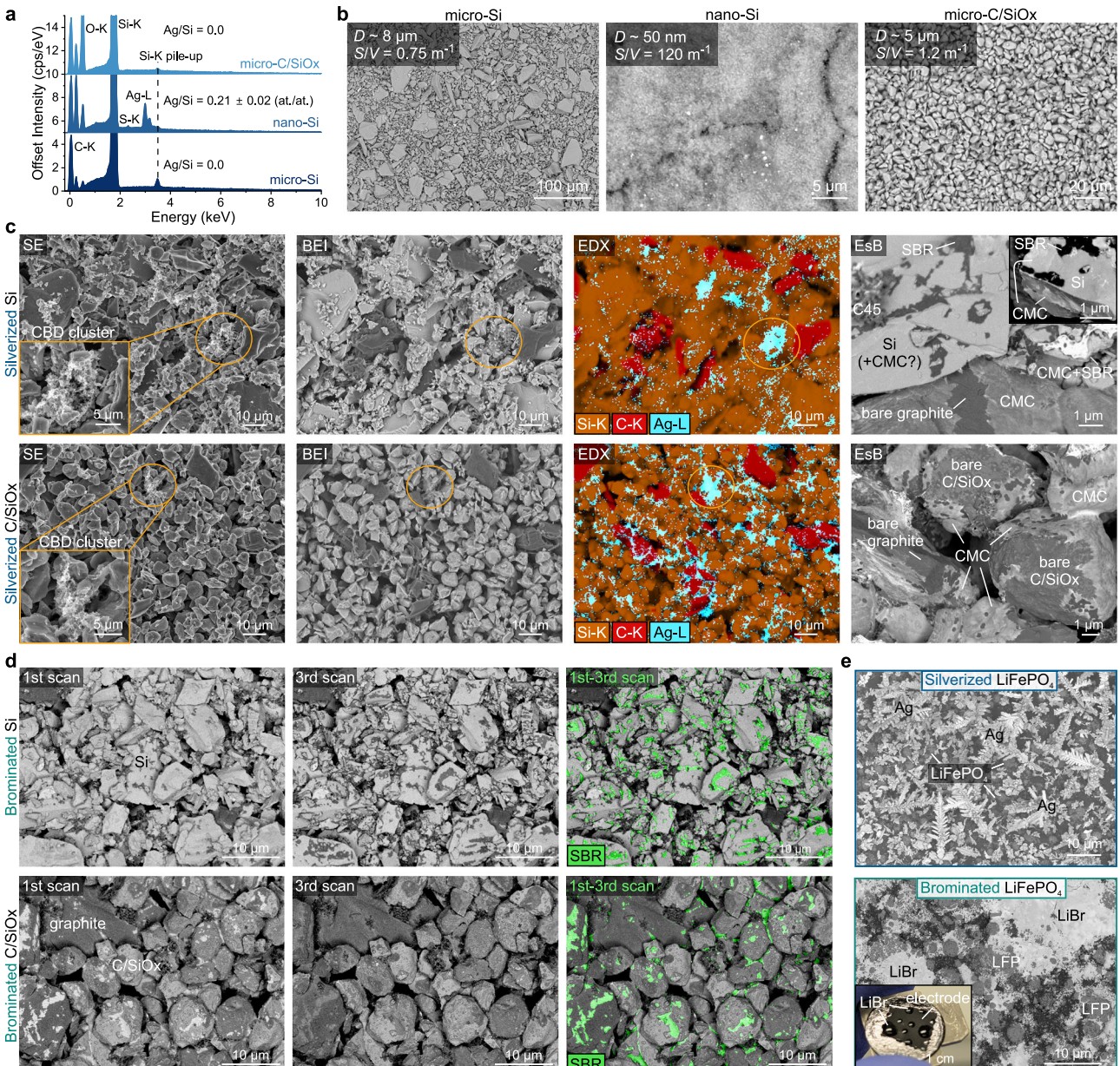

**Fig. 7 | Staining Si- and LiFePO₄-based electrodes. a** EDX spectra of silverized micro- and nano-powders of Si and C-coated SiO$_x$ with Ag/Si ratio averaged from 5 measurements per sample. **b** BEI of silverized Si and C/SiO$_x$ powders, showing average particle diameter ($D$) and surface-to-volume ratio ($S/V$). Due to the large $S/V$, the reactivity of nano-Si with Ag$^+$ will impede staining selectivity towards CMC. **c** Top down secondary electron (SE), BEI, EDX and high magnification EsB images of internal surfaces of silverized micro-Si and C/SiO$_x$ electrodes containing 20 wt% graphite, 2.3% SBR, 1.5% CMC and 1% C45. The EDX maps show that Ag was bound and accurately found in CBD carbon-binder clusters (highlighted in the SE insets). The EsB images show good contrast of the CMC nanolayer on C/SiO$_x$, but poor

contrast on Si (see CMC on graphite particles as internal contrast standard). **d** EsB images of brominated micro-Si and C/SiO$_x$ electrodes with the same composition as in **c**, together with colourised differential Br outgassing maps of SBR processed using the code in Supplementary Note 7. **e** EsB images of silverized and brominated LiFePO₄ electrodes with 1.5 wt% CMC and 2.3 wt% SBR. During silverization, Fe$^{2+}$ reduced Ag$^+$, forming large Ag dendrites. Bromination resulted in the formation of LiBr that deliquesced in air, forming liquid droplets visible in the inset photograph. Both reactions obscure the binders' detection in the LiFePO₄-based electrodes. Source data are provided as a Source Data file.

particles opens new avenues for engineering active surfaces to enhance cyclability and stability of Li-ion battery negative electrodes. During galvanostatic Li intercalation, for example, we expect that the inhomogeneously-distributed surface patches of ionically resistive CMC[76] and, in particular, thicker SBR can decrease electrochemically active surface area and cycling rate performance of graphite, also increasing local current density and local overpotential, which, in turn, could accelerate onset of localized Li metal plating and subsequent capacity drop (or failure) of the battery[74]. This could explain why excess SBR in graphitic electrodes correlates with increased internal

cell resistance, lower SEI uniformity, accelerated Li plating and degrading cell capacity, as reported recently[55]. In this context, the effect of leftover SBR polymerisation agents, uncovered here, on the cyclability of the electrodes also remains an unexplored area. On the other hand, because carboxyl-rich binders (such as CMC) stabilise the SEI layer on graphite surfaces[22,24,27], devising ways to maximise CMC coverage may also benefit cycling stability of graphitic and, in particular, volume-changing silicon electrodes[2,31,64]. The demonstrated effect of calendering on reducing CMC surface film coverage from ~90% in pristine graphite electrodes to 21% on top- and 32% within

calendered electrodes also prompts reassessing how calendering conditions affect electrochemical stability and performance of graphite-based electrodes. Considering the SEI-stabilising role of CMC, for example, the lower CMC coverage at the top electrode surface after calendering can contribute to the increased SEI growth and blocking of pores in this electrode region[74], and can partially explain why Li plating starts at the top of graphite-based electrodes, even at mild cycling conditions[27,74,75]. Therefore, the rich information provided by binder staining can shed new light on fundamental electrode processes and extend optimisation routes of Li-ion battery negative electrodes.

## Methods

### Statistics
The uncertainty (±) of values reported in this work corresponds to their expanded 95% confidence intervals estimated with the Taylor Series Method or the Monte Carlo method, considering both random and systematic uncertainties of the values.

### Chemicals
Artificial graphite (BTR S360-E3, flakes, nominal $D50 = 16$ μm), sodium carboxymethyl cellulose (degree of substitution = 0.8–0.95, Ashland), SBR (40% wt. suspension in water), carbon conductive additive (C-NERGY™ SUPER C45, Imersys), bromine liquid (99.8%, Alfa Aesar), silver nitrate (99.9999% trace metal basis, Sigma Aldrich), silicon powders (325 mesh: 99% trace metal basis, Sigma Aldrich and <50 nm: 98%, Alfa Aesar), $C/SiO_x$ powder (High Capacity Silicon Based Anode Powder, MSE Supplies), ethylene carbonate (≥ 99%, acid < 10 ppm, $H_2O$ < 10 ppm, Sigma Aldrich) and dimethyl carbonate (anhydrous, ≥99%, Sigma Aldrich) were used as received. The as-received tetrabutylammonium perchlorate (for EC analysis, ≥99.0%, Merck-Supelco) was dried in a vacuum over silica gel at room temperature for 3 days prior to bringing it into an Ar-filled glovebox to prepare the non-intercalating electrolyte for the EIS analysis. The water used for the preparation of aqueous solutions and electrode slurries was type II deionised water (0.5 μS/cm, TOC < 50 ppb). The battery-grade SBR was of a popular brand with known application in industrial Li-ion battery manufacturing, whose name is not disclosed for confidentiality.

### Electrode fabrication
Typical graphite electrodes were prepared from liquid slurries mixed using an Interionics Thinky ARE-250 planetary mixer in 35 mL polypropylene pots. First, 6 g of 1.5% wt. aqueous Na-CMC was mixed with 60 mg of C45 powder at 500 rpm for 1 min and 2000 rpm for 10 min. Then, 5.71 g of graphite powder was added and mixed at 500 rpm for 1 min and 2000 rpm for 10 min, followed by degassing at 2200 rpm for 3 min. Note that in the CBD agglomeration study (in "Correlating slurry mixing, CBD agglomeration and the electrical resistivity of graphite electrodes"), the graphite mixing was divided into two 5 min intervals with an additional 1 min 500 rpm mixing step in between, for full comparability with the modified procedure described in the same section. Finally, 0.33 g of 40% wt. aqueous SBR suspension was added and mixed at 500 rpm for 5 min. The as-obtained slurry with a total solid mass content of 49.3% was then blade-casted onto 10 μm-thick, isopropanol-wiped and dried Cu foil, using a digital adjustable applicator (QC Laboratory Equipment Ltd.), and dried in a fan convection oven for 30 min at 50 °C. The typical electrodes had a dry mass composition of 95.21 ± 0.06 wt% graphite, 2.22 ± 0.02 wt% SBR, 1.51 ± 0.01 wt% CMC and 1.05 ± 0.06 wt% C45, based on the weight of the materials used in 10 separately-made slurries. The electrodes were then cut into 20 mm-wide rectangular stripes and calendered using Sumet CA3 hot calender (150 mm drums width) operating at 80 °C, 0.2 m/min pass speed and 35 N/mm set force (corresponding to 263 N/mm of force normalised to the sample width), to produce 30% porous electrodes (standard deviation = 1%, $n = 28$).

### Bromination of electrodes
The electrodes were brominated after vacuum-drying by exposing them to vapours of $Br_2$ in an air-free setup depicted in Supplementary Fig. 1. The reaction was carried out inside a small glass reactor at 50 °C and $Br_2$ pressure of $5–8 \times 10^{-2}$ mbar for 270 s, followed by outgassing of excess $Br_2$ under active vacuum (~$2 \times 10^{-2}$ mbar). The temperature and time were chosen to ensure no condensation of $Br_2$ vapours inside the reactor and to facilitate complete reaction with SBR while minimising the processing time. $Br_2$ vapours were provided by the small $Br_2$ pool placed in a separate glass vial immersed in a small water bath (~20 °C) and connected to the pre-evacuated and heated reactor (~$2 \times 10^{-2}$ mbar, 50 °C) under static vacuum. The bromination conditions were applied without any further optimisation. After bromination, the initial samples were kept in a $N_2$-sealed vessel until EDX/BEI analysis to avoid humidity-induced hydrolysis of brominated SBR; however, our later EDX tests also indicated no significant loss of Br from the electrodes in air for at least two weeks prior to the EDX/BEI analysis.

### Staining electrodes with $Ag^+_{(aq)}$
The electrodes were silverized by immersing them in a crystallizer containing 0.5 M $AgNO_3$ for 3 min (depicted in Fig. 1a). For the CMC analysis of homogenized and bi-layered electrodes (in "Mechanism and specificity of staining reactions and Staining bi-layered electrodes", respectively), the 0.5 M $AgNO_3$ solution was freshly mixed with 1 mM Triton X-100 non-ionic surfactant prior to staining to help with penetration of the solution into all of the electrode, as discussed in detail in Supplementary Note 6 and Supplementary Fig. 8. During all silverizations, the crystallizer with $AgNO_3$-soaked electrodes was briefly placed in a desiccator under mild vacuum to facilitate penetration of $AgNO_3$ solution within the electrodes. After that time, the electrodes were taken out of the solution, washed with copious water and dried. Following silverization of the initial samples, there were visual signs of corrosion of the Cu foil and micron-sized Ag particles in random areas in the SEM images of the samples. This points to galvanic replacement of $Ag^+_{(aq)}$ with Cu, according to Eq. (1):

$$2Ag^+_{(aq)} + Cu \rightleftarrows 2Ag^0 + Cu^{2+}_{(aq)} \qquad (1)$$

To prevent it, prior to staining further electrodes, they were first carefully delaminated and transferred from the Cu foil support onto adhesive carbon SEM discs supported on a polypropylene film that was inert to Ag staining.

### Characterization
Secondary electron microscopy, BEI and EDX spectroscopy were performed using Zeiss Merlin field-emission SEM paired with Oxford Instruments (OI) XMax 150 mm² and windowless XMax 80 mm² EDX detectors operating in double detector mode, as well as Hitachi TM3030 table-top SEM paired with OI 30 mm² EDX detector equipped with SATW polymer window (used for BEI in "Correlating slurry mixing, CBD agglomeration and the electrical resistivity of graphite electrodes" and for the EDX analysis of stained individual electrode components, presented in Supplementary Information). Typical EDX acquisitions were performed at an acceleration voltage of 10 kV and a spectral resolution of 5 eV/channel using TruMap mode, following energy-calibration against a 99.9% pure Cu standard. Elemental quantification was performed using factory standards provided in the OI Aztec software version 6.1. Systematic uncertainty of the EDX quantification was verified using freshly cleaned Ag wire and dried C-coated KBr pellet with 99.9% and ≥99% trace metal-basis purity, respectively, giving 1.3 ± 1.0% and 3.2 ± 1.5% relative bias for Ag and Br quantification, respectively. Considering ~4% uncertainty due to the roughness/porosity of the graphitic samples[48], we estimate the relative combined systematic uncertainty of Ag and Br quantification at the

applied conditions and 95% confidence level as ±8.4% and ±10.4%, respectively. All EsB images were acquired at a filtering grid voltage of −513 V.

Attenuated total reflection infra-red spectroscopy (ATR-IR) was performed using a Varian Excalibur FTS 3500 FT-IR spectrometer equipped with a DTGS detector and Specac Golden Gate ATR accessory (monolithic diamond window, single bounce, 45° incidence angle, KRS-5 lenses) operating in air. For each sample, the background-corrected spectra were averaged from 256 scans acquired at 2.5 kHz speed and 4 cm⁻¹ resolution.

XPS was performed using a Thermo Scientific K-Alpha spectrometer using a monochromated Al K-alpha X-ray beam with an energy of 1486.7 eV and a voltage of 12 kV. For survey scans, a pass energy of 200 eV and a step size of 1 eV were used. For narrow scans, a pass energy of 50 eV and a step size of 0.1 eV were used. The binding energy of the spectra was corrected using the fitted saturated C-H peak at 285.0 eV. Spectra processing was done using CasaXPS 2.3.27[77].

Electrical resistivity of electrode coatings was measured using an Ossila four-point probe system equipped with spring-loaded soft contact cylindrical golden probes with 0.48 mm diameter and 1.27 mm spacing. Because the technique requires samples to be supported by non-conductive substrates, to measure uncalendared coatings, they were slurry-casted on polypropylene sheets and cut into 30×30 mm samples after drying. To measure calendered coatings, coatings were first fabricated on Cu foils and calendered to 228 μm thickness and 32% porosity at 80 °C, delaminated from Cu foils, cut to 25 × 25 mm squares and measured on top of a glass substrate. Each sample was measured 10 times, rotating it by 90° between the measurements, and the results were averaged and normalised to sample thickness (the measurements are summarised in Supplementary Table 1).

To determine the pore ionic resistivity, the electrodes were first cut into 16 mm-diameter discs and dried under active vacuum at 120 °C overnight. After drying, the electrodes were brought to an Ar-filled glovebox ($H_2O$ and $O_2$ level below 10 ppm) without breaking the vacuum and assembled in symmetric 2-electrode cells using EL-CELL PAT-Cell test cells equipped with a 260 μm-thick, 91% porous Whatman GF/A glass fibre separator. The electrolyte was 100–140 μL of 10 mM tetrabutylammonium perchlorate in 1:1 wt. ethylene carbonate/dimethyl carbonate. Care was taken to maximise the alignment of the top electrode directly above the bottom electrode by using a vacuum pen. EIS was performed using a Biologic SP-150 potentiostat in an air-conditioned room maintained at 21 ± 1 °C. The measurements were done at the DC potential equalling the open circuit potential of each cell, in the 100 kHz–10 mHz frequency range, using an AC potential amplitude of 15 mV, averaging five measures per frequency and performing three full frequency scans for each cell (with a resting period of 2 h in between the measurements). Before the first measurement, the cells were allowed to equilibrate at the open circuit potential for 8 h. The electrodes had an average loading of 14 ± 1 mg/cm² and 30 ± 3% porosity. Note that most recently, Bieneck et al. showed that the pore ionic resistance determined with symmetric-cell EIS can be strongly influenced by the spatial distribution of the double layer capacitance ($C_{dl}$) of carbon black, and its contribution to the overall $C_{dl}$ of the electrode[71]. Based on the mass composition of our electrodes and the BET surface area of the graphite and C45 carbon black (1.47 and 45 m²/g, respectively), the estimated contribution of C45 to the total $C_{dl}$ of the electrodes was ~25%. The analysis of Figure 9 in Bieneck's paper shows that for such $C_{dl}$ contribution, the pore ionic resistance of the reference electrode (with CBD concentrated close to the separator) determined with EIS may be underestimated by ~12% and that of the acetone-processed electrode (with CBD concentrated close to the current collector) may be overestimated by ~23%, implying that the actual pore ionic resistance difference between these electrodes may be ~57% instead of the apparent 40%.

## EDX data processing

All the EDX maps and derived elemental profiles were acquired and processed using Oxford Instruments Aztec 6.1 SP2 software, using signal filtering at 99% or 95% confidence threshold. The combined EDX maps in Figs. 2 and 6 were prepared by binning Ag-L and Br-L elemental maps with a factor of 2, followed by clipping 5% of the high intensity pixels and stacking with C and Cu maps using weighting settings of 1.0 (Ag or Br) and 0.4 (C and Cu) for an optimum visibility of the staining elements against the sample matrix. Note that the maps of silverized samples did not include Cu, as these samples were detached from the Cu foil prior to staining and analysis. All the EDX Br-L and Ag-L line profiles were checked for EDX geometrical artefacts by following the reference C-K signals, which should not show a significant gradient through the thickness of electrode coatings. In one sample (silverized reference electrode in the phase inversion study), a 35% positive linear gradient was present in the C-K signal from the bottom to the top of the coating. Hence, the Ag-L signal was corrected by subtracting this gradient using Eq. (2):

$$y_{corr} = y - aS(x - x_0) \qquad (2)$$

where $y$ and $y_{corr}$ is the raw and corrected Ag-L signal, respectively, $x$ is the distance (in μm), $a$ is the linear regression slope (gradient) of the C-K signal vs $x$, fitted between $x_0 = 50$ μm and $x_{end} = 137$ μm (the part of the EDX scan corresponding to the electrode coating), and $S$ is the scaling factor (the ratio of the integrated Ag-L to integrated C-K signals). The raw and corrected Ag-L and reference C-K signals are provided in the Source Data file.

## Binder quantification from BEI and EsB images

To derive the vertical distribution of binders from BEI images, the images were segmented using the Multi Otsu Threshold plugin[78] in the Fiji 2.14.0[79] package. Most of the images were segmented using three levels corresponding to the bright stained binder phase, grey graphite phase and black interparticle pore phase. In a few images with visible strong charging artefacts at the graphite particle edges, the number of levels was set to 4 to account for these artefacts, giving a smaller error in the binder phase detection.

To quantify CBD agglomerate size distribution from the BEI images in the "Correlating slurry mixing, CBD agglomeration and the electrical resistivity of graphite electrodes", the images were first processed using Smart Blur (setting 50) in CorelDRAW 2021. Then, they were thresholded in Fiji using Sauvola local threshold (radius = 100, parameter 1 = 0.2, parameter 2 = 7) followed by the particle size analysis using Fiji's default "Analyze Particles" function, considering agglomerates with at least 1 μm² surface area.

For quantification of CMC surface coverage from EsB images of silverized electrodes, for the images with high EsB noise, they were first de-noised using the default Fiji smooth filter. This was followed by selectively removing dark interparticle pores from the analysis area using Phansalkar's local threshold algorithm (radius 100, parameter 1 = 0.7, parameter 2 = 0). This was followed by applying several local and general threshold algorithms (e.g. Bernsen, Niblack, Moments, Huang) on copies of the image and averaging the results. For quantification of SBR coverage from EsB images of brominated electrodes, two EsB images acquired at the first and 3rd–5th scan (before and after Br outgassing, respectively) were first registered using the affine SIFT algorithm[80], cropped on edges, and denoised using median and non-local means filter (sigma = 5, smoothing factor = 1). The Yen threshold was applied on the nth EsB image to quantify SBR/CMC agglomerates, which were subsequently removed from the region of interest in the 1st EsB image. Phansalkar's local threshold was used to remove dark interparticle pores, followed by subtracting the nth image from the 1st image and applying Otsu or Yen threshold to quantify the surface SBR

agglomerate areas. The macro used for quantification is presented in Supplementary Note 7.

## Monte Carlo simulations

Monte Carlo simulations were performed using Casino 3.3.0.4 software[81] for a layer of Ag-CMC on a graphite substrate, using ELSEPA scattering cross-sections[82] and a modified Bethe equation by Joy and Luo for calculating stopping power[83]. The simulations were performed using $10^6$ electrons formed into a beam of 5 nm diameter and a Gaussian profile ($1.65\,nm^2$ variance), at primary electron energies of 1, 1.5, 2, 2.5 and 10 keV and a lower energy cutoff of 513 eV to reflect the EsB filter grid voltage of -513 V used in the experiments. The thickness of the Ag-CMC layer was varied between 1 and 20 nm, and the thickness of the graphite substrate was set at $1\,\mu m$ (for 1–2.5 keV simulations) or $10\,\mu m$ (for the 10 keV simulation). The stoichiometry of Ag-CMC used in the simulation was $Ag_{802}C_{7617}O_{6615}H_{10824}$, which corresponds to a Na-CMC chain with 102 glucopyranose units and a degree of substitution of 0.82, where all the Na bound to carboxylate groups was substituted with Ag (this stoichiometry was also found to have a similar atomic composition as the one determined with EDX on silverized CMC). The Ag-CMC density was set at $1.96\,g/cm^3$, based on the average reported true density of Na-CMC of $1.5\,g/cm^3$ and assuming the same molar volume of Ag-CMC as Na-CMC. The cumulative distributions of maximum penetration depth of backscattered electrons were calculated based on all the simulated electrons, while $10^4$ electrons were considered for plotting backscattered electron trajectories. These trajectories were extracted from the simulation data files into.csv files using a Python code generated with OpenAI ChatGPT-4[84] and verified by the first author for accuracy. The code is provided in Supplementary Note 8.

## Declaration of generative AI and AI-assisted technologies

This work used OpenAI ChatGPT-4[84] to assist in creating a Python code to extract electron trajectories from the Monte Carlo simulation output files generated by the Casino 3.3 Monte Carlo package[81]. The code was verified by the first author for accuracy and is provided as Supplementary Note 8.

## Data availability

All data needed to evaluate the conclusions of this paper are presented in the paper or the Supplementary Information. The raw EDX, SEM, BEI, EsB, XPS, ATR-IR, EIS and 4-point probe data generated in this study have been deposited in the Figshare database https://doi.org/10.6084/m9.figshare.28212953. Further information and requests should be directed to and will be fulfilled by the lead contacts, Stanislaw Zankowski (stanislaw.zankowski@materials.ox.ac.uk) and Patrick Grant (patrick.grant@materials.ox.ac.uk). Source data are provided with this paper.

## Code availability

The code for automated image analysis and extraction of simulated backscattered electron trajectories is provided in the Supplementary Information.

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

## Acknowledgements

We wish to acknowledge invaluable help from Dr. Phani Karamched for providing technical expertise and discussions on EDX and EsB, Dr. Colin Johnston for advising on ATR-IR, Ms. Roksana Jackowska and Prof. Emma Kendrick (University of Birmingham) for providing samples of commercial graphitic electrodes, Prof. Johannes Landesfeind (University of Agder) for advising on impedance characterization of the electrodes, Prof. Ben Davis and Prof. Jonathan Burton (University of Oxford) for discussing bromination of Na-CMC and Mr. Greg Cook and Mr. Graham Wyatt for their technical laboratory expertise. We wish to acknowledge discussions with our colleagues at the Processing of Advanced Materials Group (University of Oxford) and the academic and industrial partners of the Faraday Institution's Nextrode project. We acknowledge the use of characterisation facilities within the David Cockayne Centre for Electron Microscopy, Department of Materials, University of Oxford, alongside financial support provided by the Henry Royce Institute to SPZ and MM (Grant ref EP/R010145/1). The work of SPZ, SW, TB and PSG was supported by the Faraday Institution grant Next Generation Electrodes — Nextrode (FIRG015/FIRG066).

## Author contributions

S.P.Z. developed bromine and silver staining, performed EDX and EsB mapping, Monte Carlo simulations, electrode manufacturing studies and wrote and corrected the manuscript with valuable input from P.S.G. and T.B.; S.W. manufactured spray-coated bi-layered electrodes; T.B. assisted with electrode manufacturing, silverization and manuscript corrections; W.M.C. performed XPS analysis. M.M. assisted with EDX analysis and creating the figures; P.S.G. supervised the project, acquired the funding, corrected the manuscript and provided stimulating discussions during the project.

## Competing interests

The authors declare the following competing interests: Oxford University Innovation Limited filled provisory patent application number 2511810.0 claiming inventorship of Stanislaw P. Zankowski and Patrick S. Grant for aspects of the described methods. The application is pending revision. The remaining authors declare no competing interests.
