## [Peer Review File · Nature Communications]

Chemical staining for fundamental studies and optimization of binders in Li-ion battery negative electrodes

Corresponding Author: Dr Stanislaw Zankowski

Version 0:

Reviewer comments:

Reviewer #1

(Remarks to the Author)

The authors present an accessible approach to staining carboxymethylcellulose (CMC) and styrene butadiene rubber (SBR) binders in graphitic Li-ion electrodes with silver and bromine. This strategy enables detailed electron imaging and precise spectroscopic quantification of the binder domain and achieves a 14% reduction in electronic resistivity and a 40% decrease in electrode ionic resistance. Furthermore, staining enables unprecedented electrode-scale, high-resolution backscattered electron imaging of surprisingly complex binder hierarchies, revealing multiple types of agglomerates and hardly-ever-seen nanoscale binder films. These films completely coat graphitic surfaces in pristine electrodes but shatter into highly inhomogeneous fragments after calendaring in both research-grade and commercial electrodes, presenting new perspectives on interpreting common cycling stability and electrode performance issues. Although the study is interesting and presented results are good. I recommend the publication of this study after addressing my comments, which are given below;

1. Why are carboxymethylcellulose (CMC) and styrene butadiene rubber (SBR) specifically amenable to staining with silver and bromine? Do functional groups (e.g., carboxyl in CMC, double bonds in SBR) directly interact with these staining agents?
2. Could alternative binders (e.g., PVDF, PAA) be stained similarly, or is this method unique to CMC/SBR due to their chemical structures?
3. The authors claimed that the presented strategy achieved a 14% reduction in electronic resistivity and a 40% decrease in electrode ionic resistance. Does the staining process directly cause the reported 14% electronic resistivity and 40% ionic resistance improvements, or are they attributed to binder redistribution insights enabled by the method? Clarify causation to avoid misinterpretation.
4. Is the choice of CMC/SBR driven by their widespread use in commercial graphite anodes, making the method more applicable to real-world systems? How scalable is this staining protocol for quality control in high-throughput electrode production? Highlight time, cost, and compatibility challenges with industrial workflows (e.g., solvent safety, drying times).
5. How does the CMC/SBR ratio influence staining efficiency? Does a higher SBR content alter bromine uptake due to its hydrocarbon structure?
6. Does the mechanical flexibility of SBR or the rigidity of CMC affect the spatial distribution or adhesion of staining agents?
7. Are CMC/SBR inherently difficult to image via electron microscopy due to low conductivity or contrast, necessitating this staining method? How does staining mitigate these issues?
8. What advantages do CMC/SBR offer over other binders in balancing stainability, electrode performance, and processability?
9. Does the staining process alter the ionic/electronic conductivity of the binder domain, potentially affecting rate capability or cycle life measurements? Are there risks of silver/bromine residues introducing side reactions (e.g., redox activity) during electrochemical cycling?
10. Is this staining method applicable to next-gen anodes (e.g., silicon, lithium metal) where binders face harsher mechanical/chemical stresses? Is this staining technique compatible with operando studies of working electrodes, or is it limited to ex-situ/post-cycled samples?

Reviewer #2

(Remarks to the Author)

The authors employed silver and bromine staining to enhance the visualization of binder components in graphite anodes for lithium-ion batteries under electron imaging. This innovative approach produced exceptionally clear electron microscopy images, a result that is rarely reported in existing studies. However, the impact and significance of this finding are not adequately conveyed or discussed. Therefore, I recommend that the manuscript undergo revisions before being considered for publication or resubmission. Below are several minor issues that need to be addressed:

Figure 1(c): The authors present backscattered electron images (BEI) of silverized graphitic electrodes containing either 1.5 wt.% CMC or 2.3 wt.% SBR. What was the rationale for using different binder contents for the same silver-stained graphite electrodes? Could this variation influence the quality or interpretation of the electron imaging results?

Bromination Process: The manuscript describes the bromination of SBR in the electrodes, but it does not mention the bromination process for Na-CMC. Since both binders were brominated, it is important to include the bromination process for Na-CMC in the manuscript for consistency and clarity.

Figure Captions: In the figure captions for Figure 1 and Figure 2, the authors use two different labeling formats for subfigures [A) or (A)]. I suggest adopting a consistent labeling format throughout the manuscript to improve clarity and presentation.

Section 2.4 Alignment: At the beginning of Section 2.4, the text appears misaligned. The authors should correct the alignment to ensure proper formatting.

Viscosity and CBD Area Reduction: In Section 2.4, the authors use 3.0 wt% Na-CMC in the initial mixing step to reduce the area of the conductive carbon/binder domain (CBD). Given that the viscosity difference between 1.5 wt% and 3.0 wt% Na-CMC is substantial, the authors might want to consider using ball milling or mortar grinding for the preliminary mixing of the electrode slurry. This could further enhance the reduction of the CBD area without altering the mechanical stirring conditions. I recommend including this experiment in the manuscript as a potential improvement.

Review of Anode Active Materials: While the manuscript provides a detailed introduction to the types of binders used in lithium-ion battery anodes, it lacks a review of the anode active materials. I suggest adding a section reviewing lithium-ion battery anode active materials in the introduction, referencing relevant studies such as *Chemical Engineering Journal*, Volume 499, 2024, 156360 (ISSN 1385-8947).

Streamlining the Manuscript: The manuscript could benefit from a more streamlined structure and clearer presentation of the results. Some sections, particularly the results and discussion, are dense and would be more accessible if broken down into smaller, more digestible segments with clearer headings and subheadings.

Reviewer #3

(Remarks to the Author)

The author develops an accessible approach of staining carboxymethylcellulose (CMC) and styrene butadiene rubber (SBR) binders in graphitic Li-ion electrodes with silver and bromine, enabling detailed electron imaging and precise spectroscopic quantification of the binder domain. Notably, the staining reactions of Br₂ vapour is similar to OsO₄ staining but with significantly lower hazard, as indicated by a 217-times higher short-term safe exposure limit for Br₂ compared with OsO₄. This work can provide a guidance for slurry processing and electrode preparation of graphite anode. Therefore, this work has a chance to be published after major revise:

1. As shown in Figure 1A, the schematic diagram shows that the Ag⁺ can replace the Na⁺ in Na-CMC. It is unscientific, because the Ag⁺ tends to undergo cross-linking reactions with hydroxyl and carboxyl groups rather than substitution reactions.
2. In Figure S4, the EDX and X-ray photoemission spectroscopy (XPS) were used to quantify Br and Ag concentration. However, the measurement results of EDX and XPS are usually inaccurate when compared with Inductively Coupled Plasma (ICP). It is better to provide the ICP measurement of Br and Ag concentration.
3. As illustrated in Figure 5A, the author claims that the more concentrated CMC solution in the first mixing step might improve the final C45 dispersion in the electrode. What's the specific mechanism? Generally, the concentrated binders solutions are not conducive to dispersion of electrode materials.
4. It is better to provide a optimized concentration of CMC solution in the first mixing step, which is important for the slurry processing in Li-ion battery industry.
5. In Figure 6, when the polymer solution (e.g. PVDF in N-methyl pyrrolidone) is immersed in a non-solvent (e.g. water), the adhesion of polymer slurry on Cu foil will greatly decrease and even detach from the Cu foil surface. This phenomenon will increase the contact resistance of electrode materials and further decrease the electrochemical performance of Li-ion battery. However, the author reports a opposite result by attributing to the reduced pore resistance? I think that it is hard to convince. The author should provide more discussions and comparisons in this section by supplementing experiment.
6. As we know, the Br₂ vapours will corrode the Cu foil, CMC and SBR during bromination of electrodes at high temperature, which is very detrimental to the electrochemical performance of Li-ion battery. Therefore, how to avoid this phenomenon during practical application?
7. The article contains some technical terms and abbreviations (such as EDX, BEI, EsB, etc.). While these are helpful for concise expression, they may pose comprehension barriers for non-specialist readers. It is suggested that the authors provide detailed explanations when these terms first appear.
8. Although the authors have explained the mechanism of the staining reactions, the descriptions of the optimization processes for some key parameters are not detailed enough. For example, the rationale behind the selection of reaction temperature, time, and bromine vapor concentration in the bromination process has not been fully elaborated. It is suggested that the authors provide more detailed information on these optimization processes to help readers better understand and reproduce the method.
9. While the authors have observed the complex distribution patterns of binders within the electrodes, the in-depth analysis of how these patterns affect the electrochemical performance of the electrodes is not sufficiently thorough. For example,

regarding the role of nanoscale binder films during charge-discharge processes and the impact of non-uniform binder distribution on battery cycling stability, it is recommended that the authors conduct a more in-depth investigation in conjunction with the results of electrochemical testing.

10. While the article mentions the advantages of CMC and SBR binders over traditional PVDF binders in terms of staining, the comparative analysis with other potential staining methods may not be comprehensive enough. It is suggested that the authors include a comparison with other staining methods in the discussion section or elaborate on the universality of this staining approach.

11. In presenting key data, such as quantitative analysis of binder distribution, the authors provided EDX and BEI images for only part of the electrodes, which may not fully represent the entire electrode. It is recommended to increase the image coverage to more comprehensively show the overall characteristics of the electrodes. For example, larger - area images could be provided, or multiple images taken at different positions and stitched together to ensure data integrity and representativeness.

Reviewer #4

(Remarks to the Author)

In this manuscript, authors suggested a new method that staining carboxymethylcellulose (CMC) and styrene butadiene rubber (SBR) binders in graphitic Li-ion electrodes with silver and bromine, enabling detailed electron imaging and precise spectroscopic quantification of the binder domain. By using this method, the authors could achieve the suppression of binder migration during high-temperature electrode drying, and a 40% decrease in electrode ionic resistance for unprecedented electrode-scale, high-resolution backscattered electron imaging of surprisingly complex binder hierarchies, revealing multiple types of agglomerates and hardly-ever-seen nanoscale binder films. In conclusion, the rich information provided by binder staining can shed new light on fundamental electrode processes and extend optimization routes of negative Li-ion electrodes. Therefore, I suggest this manuscript to be published in journal "Nature Communications " after minor revisions according to the following aspects:

1. The authors conducted special analysis for SE, EDX, BEI. If possible, please provide XPS mapping data for not only elemental analysis but also chemical state analysis.
2. In Fig4e, the authors provided graphite surface coverage data to confirm binder distribution. Detailed process to get surface coverage data should be described.
3. In this manuscript, the authors only considered graphite as an anode materials. Please provide possibility to use this method to the other electrode materials.
4. In Fig3a, the authors showed images of internal electrode surface exposed by scotch tape peel-off and subsequent silverization, imaged at different electron beam voltages with surface sensitivity. How does the surface sensitivity work with electron beam voltages?
5. The authors claimed that they could reduce 14% of electronic resistivity and decrease 40% of ionic resistance. What is possible way to increase each value?

Version 1:

Reviewer comments:

Reviewer #2

(Remarks to the Author)

This study, for the first time, achieves nanoscale visualization of aqueous binders in Li-ion battery electrodes through chemical staining coupled with high-resolution imaging, unveils their microstructural evolution mechanisms, and offers a fresh paradigm for electrode performance optimization and manufacturing control. The authors have provided a thorough response to the reviewers' comments and expanded the dataset; the paper can be accepted for publication.

I have reviewed the authors' responses and find that they have satisfactorily addressed all the reviewers' comments. I have no further comments.

Reviewer #3

(Remarks to the Author)

The revised manuscripts can be published by Nature communications.

The revised manuscript has addressed referee #1's comments, and therefore can be accepted by Nature Communications.

NCOMMS-25-04022 **Chemical staining – a toolbox for fundamental studies and practical optimization of binders in negative Li-ion electrodes**

Dear Reviewers,

Thank you for your time and valuable suggestions on our manuscript *Chemical staining – a toolbox for fundamental studies and practical optimization of binders in negative Li-ion electrodes*. We have thoroughly addressed your comments, including performing a number of new experiments as requested, and modified our manuscript accordingly. Because of the article size limit for *Nature Communications*, we have kept modifications to the main article concise without comprising clarity. Below, we reproduce each of the comments together with our response, and the related changes in the manuscript are highlighted in yellow.

Reviewer #1 (Remarks to the Author):

The authors present an accessible approach to staining carboxymethylcellulose (CMC) and styrene butadiene rubber (SBR) binders in graphitic Li-ion electrodes with silver and bromine. This strategy enables detailed electron imaging and precise spectroscopic quantification of the binder domain and achieves a 14% reduction in electronic resistivity and a 40% decrease in electrode ionic resistance. Furthermore, staining enables unprecedented electrode-scale, high-resolution backscattered electron imaging of surprisingly complex binder hierarchies, revealing multiple types of agglomerates and hardly-ever-seen nanoscale binder films. These films completely coat graphitic surfaces in pristine electrodes but shatter into highly inhomogeneous fragments after calendaring in both research-grade and commercial electrodes, presenting new perspectives on interpreting common cycling stability and electrode performance issues. Although the study is interesting and presented results are good. I recommend the publication of this study after addressing my comments, which are given below;

1. Why are carboxymethylcellulose (CMC) and styrene butadiene rubber (SBR) specifically amenable to staining with silver and bromine? Do functional groups (e.g., carboxyl in CMC, double bonds in SBR) directly interact with these staining agents?

In the first paragraph of Results (2.1) we explain the mechanism of CMC staining with Ag and SBR staining with Br. To improve clarity we have slightly modified the first sentence and we provide the key chemical reaction equations in Fig. 1a and 1b, showing bonding of Ag^+ to the $-\text{COO}^-$ groups of the CMC, and of Br_2 to the $\text{C}=\text{C}$ groups of SBR:

Our concept of binder staining relies on exposing electrode samples to chemical agents that specifically attach to the functional groups present within the binder structure but not in the other components of the electrode, as shown schematically in Figure 1. Immersing graphite electrodes in aqueous AgNO_3 solution leads to Ag^+ ions binding to the carboxyl groups in CMC, forming a water-insoluble complex depicted in Fig. 1a. On the other hand, exposing the electrodes to Br_2 vapour brominates the aliphatic sp^2 carbons of SBR, as depicted in Fig 1b.

We also direct the readers to Supporting Figure S2 where we provide the ATR-IR (now updated with EDX) data confirming changes in the carboxylate bands of the CMC upon silverization, and in the $\text{C}=\text{C}$ peaks of SBR upon bromination. We have now also added XPS C 1s, Ag 3d and Br 3d core level analysis (Supporting Note S2 and Supporting Figure S4) that also support these findings. These observations confirm the direct interaction of the staining agents with the carboxylate groups of CMC and aliphatic $\text{C}=\text{C}$ groups of SBR.

The mechanism of both reactions was confirmed using attenuated-total reflection infrared spectroscopy (ATR-IR), X-ray photoemission spectroscopy (XPS) and energy-dispersive X-ray spectroscopy (EDX), showing diminished aliphatic $\text{C}=\text{C}$ peaks in brominated SBR, a strong shift of $-\text{COO}^-$ peaks in silverized CMC and substitution of Ag for Na, as discussed in Supporting Notes S1-S2 and Supporting Figures S2-S4.

We also provide extended discussion on the mechanism of the staining reactions in Supporting Note S1:

The mechanism of staining reactions was first assessed by ATR-IR. Figure S2 shows ATR-IR spectra of pristine and freshly brominated SBR (Fig. S2A), as well as spectra of pristine CMC and CMC after exposure to Ag^+ (Fig S2B). As expected, the spectra of SBR before and after bromination indicated diminishing intensity of aliphatic $\text{C}=\text{C}$ peaks, particularly the trans peak at 967 cm^{-1} , vinyl peaks at 909 , 995 and 1640 cm^{-1} and cis peak at 1403 cm^{-1} , with the appearance of a broad shoulder around 1225 cm^{-1} , which could be ascribed to the C-H wagging vibrations within the $\text{CH}_2\text{-Br}$ group.¹⁻³ Additionally, the appearance of the carbonyl peak at 1717 cm^{-1} and, potentially, ester peak at 1240 cm^{-1} can be indicative of the partial decomposition of C-Br bonds in air.⁴ This reaction did not occur at high enough rate to be of concern for EDX mapping of binders, with no significant lowering of Br

concentration in samples exposed to air for up to a year. Bromination of a thick and compact SBR film resulted in bromination of the top 1-2 μm part of the film, as evident from the backscattered electron image of the sample cross section (Figure S3).

The ATR spectra of CMC in Fig. S2B showed four distinctive regions: 3700-3000 cm^{-1} related to vibrations of OH groups (linked to pyranose rings of CMC, physisorbed water and intra- and intermolecular hydrogen bonds), 3000-2800 cm^{-1} related to vibrations of CH and CH_2 groups, 1800-1200 cm^{-1} related to vibrations of carboxylate groups, and $< 1200 \text{ cm}^{-1}$ related to vibrations of pyranose rings and glycosylic linkages between them.⁵⁻⁷ The spectra showed strong interactions between the complexing metal ions and carboxylate groups of CMC upon staining with Ag^+ , as evident by the downshifting of the peaks at 1588 cm^{-1} and 1414 cm^{-1} associated with asymmetric and symmetric vibrations of COO^- groups. The separation between these two peaks could also be related to the structure of the carboxylate-metal complex.^{8,9} The peak separation of $174 \pm 2 \text{ cm}^{-1}$ for Na-CMC and $154 \pm 4 \text{ cm}^{-1}$ for Ag-CMC indicated that Ag binding changed the metal-CMC coordination from bidentate bridging to bidentate chelating geometry.^{8,9} Although there were no significant changes in the hydroxyl region between the samples, a fuller analysis of this region may be obstructed by the presence of adsorbed water.

2. Could alternative binders (e.g., PVDF, PAA) be stained similarly, or is this method unique to CMC/SBR due to their chemical structures?

Yes, our investigations of sodium alginate and polyacrylate showed successful binding of Ag to these polymers, as mentioned in Section 2.1:

The high affinity of Ag^+ for carboxylate groups was not limited to CMC – silverization was also successfully applied to carboxyl-rich sodium alginate and sodium polyacrylate – two other widely researched aqueous binders with a superior performance in e.g. silicon anodes^{31,34,40} (Supporting Fig. S3).

To strengthen further, we have now added a new result on EsB imaging of PAA in graphitic electrodes in Section 2.3 (*EsB imaging of local binder morphology*):

The thin layer morphology could also be observed in EsB images of silverized electrodes with an analogous polyacrylate binder (Supporting Figure S15).

We have also clarified this aspect further in the comment to the *Limitations* discussion:

Figure S15. EsB and secondary electron (SE) imaging of internal surface of a silverized electrode consisting of 1.5 wt.% polyacrylic acid (PAA), 95.2 wt.% graphite, 2.3 wt.% SBR and 1 wt.% C45. The images were taken at 10 kV (top left) or 1 kV (all the other images) of primary electron beam voltage and show that PAA adapted the same type of morphologies on graphite as CMC. The high magnification images at the bottom show fractured graphite particle and estimated PAA layer thickness, based on 4 (EsB) and 5 (SE) measurements along the dotted lines. The average PAA layer thickness is similar to the one of CMC in an analogous electrode (Figure S14).

Also, because Ag^+/Br_2 staining targets carboxylate and aliphatic C=C groups in the binders, processes that alter these groups beforehand (e.g. drying-induced cross-linking) may reduce Ag/Br uptake (which, however, could be exploited to quantify binder cross-linking). Consequently, bromination and silverization are not suitable for staining e.g. PVDF or PTFE binders (that lack these functional groups) but can be suitable for other binders, such as the carboxyl-rich alginates and polyacrylates (as shown in Supporting Figure S15). However, the accuracy of spatial mapping these two binders in electrodes remains to be assessed.

3. The authors claimed that the presented strategy achieved a 14% reduction in electronic resistivity and a 40% decrease in electrode ionic resistance. Does the staining process directly cause the reported 14% electronic resistivity and 40% ionic resistance improvements, or are they attributed to binder redistribution insights enabled by the method? Clarify causation to avoid misinterpretation.

Thank you for this comment – based on this and a few other points raised by other reviewers, we realized there was some confusion about the staining concept. The staining is performed on electrode samples cut out from the electrode sheets *only for binder analysis* – it is not applied on the part of electrode sheets that are used for electrochemical testing (nor for making batteries). Thus, all reported differences in properties (e.g. electronic/ionic conductivity) of differently processed electrodes are a result of the processing itself (e.g. preparation of slurries from dilute vs. concentrated CMC, or drying with or without phase inversion) – and not the staining. We now clarify at the end of the paragraph introducing the staining concept (section 2.1):

The high affinity of Ag⁺ for carboxylate groups is not limited to CMC – silverization was also successfully applied to carboxyl-rich sodium alginate and sodium polyacrylate – two other widely researched aqueous binders with a superior performance in e.g. silicon anodes^{31,39} (Supporting Figure S3). **Note that because of the reactivity of Ag⁺_(aq) and Br₂, staining and binder mapping should be applied on separate electrode areas to those used in electrochemical or other testing.**

We have also clarified it in Section 2.5 (CBD agglomeration study):

Furthermore, four-point probe measurements **of not-stained electrodes** showed [...]

And Section 2.6 (phase inversion study):

To explore effects on pore ionic resistance, **we investigated unstained electrodes** using electrochemical impedance spectroscopy (EIS) in a non-intercalating electrolyte developed by Landesfeind *et al.*^{3,54}

We also added a note to the *Limitations* section:

For the same reason, silverization requires impractical detachment of any Cu current collector prior to staining, **and the likely alteration of the electrode's electrochemistry by Ag⁺ and Br₂ staining may restrict its application, for example, in operando studies or on-line manufacturing quality control.**

4. Is the choice of CMC/SBR driven by their widespread use in commercial graphite anodes, making the method more applicable to real-world systems? How scalable is this staining protocol for quality control in high-throughput electrode production? Highlight time, cost, and compatibility challenges with industrial workflows (e.g., solvent safety, drying times).

Yes, we focused on CMC/SBR because of their widespread adaptation in commercial graphitic electrodes, and because of the lack of convenient methods for analysing their spatial distribution within these electrodes. We explain in the Introduction:

Since the early 2000's, the water-soluble sodium salt of carboxymethyl cellulose (Na-CMC) blended with emulsified styrene butadiene rubber (SBR) has become the now-standard binder mixture used in industrial graphite and graphite/silicon negative Li-ion electrodes,³⁰ and is also being explored for next-generation Si anodes³¹ and water-processable LiFePO₄ cathodes.^{32,33} Being dispersible in water rather than toxic N-methyl pyrrolidone (NMP), these binders considerably lowered cost and complexity of anode manufacturing, and improved anode's cycling stability.^{1,34} Contrary to the older-generation polyvinylidene fluoride (PVDF) binder, however, SBR and CMC lack distinct, covalently bound elements that are traceable by readily accessible methods such as energy-dispersive X-ray spectroscopy (EDX).

Please note, that in our approach we did not choose CMC and SBR as the binders applicable for the staining approach (i.e., we did not first develop the staining methods, and then chose the binders that it could be applied to). Instead, we first realized that there are no practical methods for mapping CMC/SBR (the two most common binders in commercial Li-ion anodes), and based on that knowledge, we then designed these staining methods to analyse particularly these (and other analogous) binders.

As for the scalability and industrial application, we now include a note on the large area cross-sectioning preparation in the *Limitations* section:

To minimize geometrical artifacts, the cross-sectional binder analysis requires lengthy sample plasma cross-sectioning (~8-12 hours for 100 µm-thick electrodes), which might be shortened using laser pre-cutting or resin impregnation and microtomy.

Following that, we have included brief discussion in the Discussion and Summary (Section 3):

Staining combined with BEI, in particular, is simple and attractive for binder-informed optimization and off-line quality control of electrodes in industrial settings, considering availability

of BEI even in table-top SEMs. The cost of implementing these methods comprises capital and operational cost of EDX/BEI-equipped SEM, while the staining itself utilizes simple equipment and very small quantities of relatively cheap and re-usable chemicals (here, 0.3 g AgNO₃ or 0.6 g Br₂ per batch of up to 5 samples). With optimization of sample-cross sectioning, the complete staining and through-thickness EDX analysis time may be reduced from 15 to 5 h/sample and down to 3 h/sample for BEI analysis. Note that the staining and analysis not requiring cross-sectioning, such as top-down CBD agglomeration check with BEI, can be already performed at < 1 h/sample.

Note the only solvent used in the processing was water. We have also only focused on the cross-sectioning time as it was the limiting factor for the total processing time. The drying time after silverization (30 min) was insignificant compared to sample cross-sectioning time (10-12 h for 100 um cross-sections).

5. How does the CMC/SBR ratio influence staining efficiency? Does a higher SBR content alter bromine uptake due to its hydrocarbon structure?

We have performed an additional staining campaign on newly-made electrodes with different CMC+SBR ratios and a total binder mix of 4% wt. Multiple areas of delaminated electrodes with a total area of ~20 cm²/coating were stained with Ag⁺ or Br₂ using the protocols already described, and the amount of Ag or Br was analysed with EDX. To remove any influence of binder migration and to increase representativeness of the average staining efficiency, prior to EDX analysis, we homogenized each batch of samples by pulverizing them in a mortar, followed by mounting the powders onto 12 mm-diameter SEM sticky carbon discs. EDX analysis was conducted on at least 5×1.6 mm² random areas per sample. We additionally verified the accuracy of our EDX Ag and Br factory-standardized quantification routine using freshly cleaned >99.9% pure Ag wire and a >99% pure KBr pellet, which gave very small relative Ag and Br quantification biases of only 1.3% and 3.2%, respectively.

Figure 1. Mechanism and selectivity of the staining methods. **a** Photograph of a supported fragment of a graphite electrode immersed in aqueous AgNO_3 during silverization, together with a reaction scheme of Ag^+ binding to Na-CMC. **b** Photograph of supported electrode fragments during bromination, together with a reaction scheme of Br binding to SBR (primary reaction) and Na-CMC (side reaction). **c** Backscattered electron imaging (BEI) of silverized and homogenized graphitic electrodes containing either 4 wt.% CMC or 4 wt.% SBR. **d** BEI of brominated and homogenized CMC- and SBR-only graphitic electrodes. **e** Atomic % of Ag and Br in stained and homogenized electrodes containing 4 wt.% of CMC+SBR mixture with different CMC:SBR ratios, determined with EDX at 10 kV. Each point represents Ag/Br content averaged from at least five $1.46 \times 1.09 \text{ mm}$ area measurements. **f** EDX Br content in brominated electrodes during continuous irradiation with SEM/EDX electrons, normalized to the Br content averaged over the first 10 seconds of the experiment. For each sample, $152 \times 114 \mu\text{m}$ area was exposed to 5 kV, 2 nA electron beam current and EDX spectra were continuously acquired every 3.5 seconds. **g** Final normalized Br content in the electrode fragments, averaged between 170-200 seconds of the experiment in **f**. The error bars and shaded areas in **e-g** represent 95% confidence intervals of the average values and linear regression fits, respectively. The samples in **c-e** were silverized in 0.5M AgNO_3 with an addition of 1 mM Triton X-100 surfactant, as explained in the Methods. The contrast of BEI images was equalized for comparable visibility of graphite and stained binders.

The new results showed a linearly increasing amount of Ag with the amount of CMC in the coatings, and similarly, a linearly increasing amount of Br with the amount of SBR in the coatings, with R^2 between 0.978-0.994. The two formulations containing either only 4% wt. CMC or 4% wt. SBR additionally allowed us to quantify selectivity. We found $\sim 16\times$ larger amount of Ag binding per unit mass of CMC compared with SBR, and $\sim 7\times$ larger amount of Br binding per unit mass of SBR compared with CMC in the electrodes. These values are also similar to the values derived from the slopes and intercepts of a linear fits to data over the composition range ($16\times$ and $5.5\times$ selectivity, respectively). *This confirmed that within the analysed binder content (which is representative to the typical binder content in commercial electrodes), staining efficiency/selectivity is largely independent on the CMC:SBR ratio.*

Using the set of brominated electrode coatings with known, linearly changing SBR content, we additionally quantified the correlation between the amount of SBR in the electrodes and the degree of electron-induced Br outgassing from Br-SBR after brominating the samples (which we previously quantified only in two samples containing either 1.5% CMC or 2.3% SBR). We found a linearly increasing degree of Br outgassing with the increasing SBR content ($R^2 = 0.944$), showing that Br outgassing can be used for selective quantification of SBR content in the samples containing CMC.

This new analysis is more robust than the previous analysis of only two electrodes with either 1.5 wt.% CMC or 2.3% wt. SBR so we have substituted the new results for the previous ones – updating Figure 1 and the corresponding discussion and moving the previous results to the Supporting Information.

6. Does the mechanical flexibility of SBR or the rigidity of CMC affect the spatial distribution or adhesion of staining agents?

Because both bromination and silverization are performed using highly mobile reagents (Br_2 vapours and Ag^+ ions dissolved in water), we do not expect mechanical properties of CMC and SBR to affect spatial distribution and adhesion.

7. Are CMC/SBR inherently difficult to image via electron microscopy due to low conductivity or contrast, necessitating this staining method? How does staining mitigate these issues?

The inherent difficulty of observing CMC and SBR in electrodes by electron microscopy is due to a combination of factors, depending on the mode employed:

- amorphous structure
- low fraction
- nanometer dimensions of the binders (in some areas)
- degradation by the electron beam
- chemical composition highly similar to graphite
- insufficient surface and chemical sensitivity of backscattered electron detectors

Particularly in the case of secondary electron imaging in the SEM, distinguishing between materials requires them to have different morphology, but binder morphology largely templates the graphite.

For imaging with chemically-sensitive back-scattered electron imaging BEI, both SBR and CMC are chemically highly similar to the graphite but staining them with much higher atomic number Br and Ag, as we show, readily allows for BEI and EDX.

We have now condensed this information in the paragraph in the Introduction:

Contrary to the previous-generation polyvinylidene fluoride (PVDF) binder, however, SBR and CMC lack distinct, covalently bound elements that are traceable by readily accessible methods such as energy-dispersive X-ray spectroscopy (EDX) or backscattered electron imaging (BEI). Lacking a well-defined morphology, binders are also difficult to distinguish with scanning electron microscopy (SEM) techniques.

8. What advantages do CMC/SBR offer over other binders in balancing stainability, electrode performance, and processability?

We have now added a short note on that to the introduction:

Being dispersible in water rather than toxic N-methyl pyrrolidone (NMP), these binders have lowered the cost and complexity of anode manufacturing, additionally improving cycling stability.^{1,34}

9. Does the staining process alter the ionic/electronic conductivity of the binder domain, potentially affecting rate capability or cycle life measurements? Are there risks of

silver/bromine residues introducing side reactions (e.g., redox activity) during electrochemical cycling?

As detailed in Answer 3 above, staining is not applied to electrodes subject to any other testing than binder mapping, so it does not affect cyclability of actual electrodes.

10. Is this staining method applicable to next-gen anodes (e.g., silicon, lithium metal) where binders face harsher mechanical/chemical stresses? Is this staining technique compatible with operando studies of working electrodes, or is it limited to ex-situ/post-cycled samples?

We have now fabricated new CMC/SBR-based electrodes made of SiO_x/graphite, micro-Si/graphite and LiFePO₄, stained them with Ag and Br and investigated by EsB/BEI/EDX. We also verified reactivity of the AgNO₃ with nano- and micropowders of Si and C-coated SiO_x. We have included the results in the new Figure 7 and *Limitations* section:

First, the staining methods, particularly the liquid-phase silverization, are less applicable to electrodes containing easily-oxidizable materials (e.g. metallic, LiFePO₄ or nano-Si particles), as these can rapidly undergo redox reactions with the staining elements, complicating image interpretation. Micro-Si or materials with an inert, stable surface passivation layer can be, however, suitable. This is shown in Figure 7 that demonstrates successful staining on Si and C-coated SiO_x microparticle-based electrodes but not on the more reactive LiFePO₄ or nano-Si. For the same reason, silverization requires detachment of any Cu current collector prior to staining, which is often impractical in an industrial setting, and the likely alteration of the electrode's electrochemistry by Ag⁺ and Br₂ staining may restrict its application, for example, in operando studies or on-line quality control.

The above comment also covers that the method will not be compatible with reactive and/or water sensitive metals such as Li metal.

As for the operando studies, we do not recommend performing them on stained electrodes due to the side-effects of the staining (current collector oxidation by Br₂, introduction of electrochemically-active Ag⁺ to the system), which will likely alter original electrochemical behaviour of the electrodes, making such studies not representative for the actual behaviour of Li-ion cells. For the same reasons, we do not perform staining on electrodes subject to EC cycling, as explained in the *Limitations* note above and in the comment to Question 1.3.

Figure 7. Staining Si- and LiFePO₄-based electrodes. **a** EDX spectra of silverized micro- and nano-powders of Si and C-coated SiO_x with Ag/Si ratio averaged from 5 measurements per sample. **b** BEI of silverized Si and C/SiO_x powders, showing average particle diameter (D) and surface-to-volume ratio (S/V). Due to the large S/V, the reactivity of nano-Si with Ag⁺ will impede staining selectivity towards CMC. **c** Top down secondary electron (SE), BEI, EDX and high magnification EsB images of internal surfaces of silverized micro-Si and C/SiO_x electrodes containing 20 wt.% graphite, 2.3% SBR, 1.5% CMC and 1% C45. The EDX maps show that Ag was bound and accurately found in CBD carbon-binder clusters (highlighted in the SE insets). The EsB images show good contrast of CMC nanolayer on C/SiO_x but poor on Si (see CMC on graphite particles as internal contrast standard). **d** EsB images of brominated micro-Si and C/SiO_x electrodes with the same composition as in **c**, together with colored differential Br outgassing maps of SBR processed using Supplementary Code S1. **e** EsB images of silverized and brominated LiFePO₄ electrodes with 1.5 wt.% CMC and 2.3 wt.% SBR. During silverization, Fe²⁺ reduced Ag⁺ forming large Ag dendrites. Bromination resulted in formation of LiBr that deliquesced in air forming liquid droplets visible in the inset photograph. Both reactions obscure binders' detection in the LiFePO₄-based electrodes.

Reviewer #2 (Remarks to the Author):

The authors employed silver and bromine staining to enhance the visualization of binder components in graphite anodes for lithium-ion batteries under electron imaging. This innovative approach produced exceptionally clear electron microscopy images, a result that is rarely reported in existing studies. However, the impact and significance of this finding are not adequately conveyed or discussed. Therefore, I recommend that the manuscript undergo revisions before being considered for publication or resubmission. Below are several minor issues that need to be addressed:

1. *Figure 1(c): The authors present backscattered electron images (BEI) of silverized graphitic electrodes containing either 1.5 wt.% CMC or 2.3 wt.% SBR. What was the rationale for using different binder contents for the same silver-stained graphite electrodes? Could this variation influence the quality or interpretation of the electron imaging results?*

We originally chosen 1.5 wt.% CMC and 2.3 wt.% SBR to mimic industrial practice but we have now investigated staining of electrodes with both equal binder contents, and a wider range of CMC:SBR ratios. We have compared the staining efficiency EDX data of the new electrodes (with 4% binders) to the original ones, and the Ag / Br quantities normalized to different binder content were very similar at the same analysis conditions (which is a consequence of strongly linear dependency between Ag/Br content and CMC/SBR content, as established in the new experiments described in the updated Figure 1). In updated Fig. 1 (also attached above, Page 8), we have now provided BEI images of the 4% CMC or SBR electrodes, in which the binder morphologies are analogous to the previous BEI images of 1.5% CMC / 2.3% SBR electrodes.

2. *Bromination Process: The manuscript describes the bromination of SBR in the electrodes, but it does not mention the bromination process for Na-CMC. Since both binders were brominated, it is important to include the bromination process for Na-CMC in the manuscript for consistency and clarity.*

We have had new, detailed discussion with research chemists with relevant experience and we now provide the potential reaction equation in Fig. 1a: M^+ -assisted Hunsdiecker bromination of carboxylic groups in the Na-CMC followed by air-hydrolysis of the brominated organic product and leaving NaBr as the predominant Br-containing compound in brominated Na-CMC, as now explained in Section 2.1:

The small amount of Br binding to CMC was due to surface oxidation of Na-CMC by Br₂ and binding of Br to Na (see exemplary Hunsdiecker reaction scheme in Fig. 1b).⁴²

Further investigation would require detailed (e.g. NMR) analysis that is outside the scope of this work - hence, we note *exemplary reaction scheme*.

3. *Figure Captions: In the figure captions for Figure 1 and Figure 2, the authors use two different labelling formats for subfigures [A] or (A)]. I suggest adopting a consistent labelling format throughout the manuscript to improve clarity and presentation.*

We have now changed all the figure captioning formats to follow the standard adopted across Nature publications

4. *Section 2.4 Alignment: At the beginning of Section 2.4, the text appears misaligned. The authors should correct the alignment to ensure proper formatting.*

We have checked now to ensure the alignment is correct.

5. *Viscosity and CBD Area Reduction: In Section 2.4, the authors use 3.0 wt% Na-CMC in the initial mixing step to reduce the area of the conductive carbon/binder domain (CBD). Given that the viscosity difference between 1.5 wt% and 3.0 wt% Na-CMC is substantial, the authors might want to consider using ball milling or mortar grinding for the preliminary mixing of the electrode slurry. This could further enhance the reduction of the CBD area without altering the mechanical stirring conditions. I recommend including this experiment in the manuscript as a potential improvement.*

Following this insightful suggestion, we fabricated an electrode using the standard method (with initial CMC concentration of 1.5 wt.%) but with addition of zirconia balls to the mixing pot to see if they could provide better CBD dispersion. Milling balls provided only a moderate decrease in CBD clustering, from 8.5 clusters/mm² (reference electrode) to 4.8 clusters/mm², but still quite far from 0.4 clusters/mm² in the electrode prepared from the modified slurry with initially-concentrated (3%) CMC. We have now described this in the same section:

Figure 5B shows a BEI image of a modified electrode after bromination, confirming the disappearance of large CBD clusters, which is also evident in the histogram in Fig. 5C.

Interestingly, a similar level of CBD de-agglomeration could not be achieved using zirconia balls during mixing of the standard 1.5 wt. % CMC slurry, indicating stronger influence of the initial CMC concentration on CBD dispersion (Supporting Figure S17).

Figure S15. Comparison of carbon-binder domain (CBD) agglomeration in electrodes made from standard (1.5 wt. % CMC initial) slurry, the same type of slurry with an addition of two zirconia balls to the mixing pot, or the modified (3.0 wt.% CMC initial) slurry. (A) Representative color-enhanced backscattered electron images (BEI) of the top electrode surfaces, showing CBD as the bright blue phase; (B) Histograms showing size distribution of the CBD clusters in the three types of electrodes. BEI image segmentation and CBD clustering analysis was done according to Supporting Procedure S2.

Following this and the similar Question 4.10, we also included it in the comment:

Future staining-assisted electrode optimization could investigate alternative pre-mixing methods of thinner, lower CMC-content slurries, or substituting C45 nanoparticles with carbon nanotubes or graphene for a further resistivity decrease.⁶

6. Review of Anode Active Materials: While the manuscript provides a detailed introduction to the types of binders used in lithium-ion battery anodes, it lacks a review of the anode active materials. I suggest adding a section reviewing lithium-ion battery anode active materials in the introduction, referencing relevant studies such as *Chemical Engineering Journal*, Volume 499, 2024, 156360 (ISSN 1385-8947).

Thank you for directing us to this article. We are not sure if the above reference is correct, as the one we found using the above data (“Robust self-healing ion-conductive interlocking

dual-network binder based on gradient dynamic bonding for advanced SiO anodes in Li-Ion batteries” by Weng *et al.*) is not a review of anode active materials but describes a functionalized PAA binder for SiO anodes. Still, it is an interesting demonstration of using a functionalized cross-linked and self-healing PAA binder in SiO anodes, thus we have included it in the Introduction as one of the references on PAA binders for silicon-based anodes:

The high affinity of Ag⁺ for carboxylate groups is not limited to CMC – silverization was also successfully applied to carboxyl-rich sodium alginate and sodium polyacrylate – two other widely researched aqueous binders with a superior performance in e.g. silicon anodes^{31,33,40} (Supporting Fig. S3).

7. Streamlining the Manuscript: The manuscript could benefit from a more streamlined structure and clearer presentation of the results. Some sections, particularly the results and discussion, are dense and would be more accessible if broken down into smaller, more digestible segments with clearer headings and subheadings.

We have now additionally split Results sub-section 2.3 (describing EsB imaging of binders on active particle surfaces) into two parts (2.3 describing EsB detection of the binders and Monte Carlo estimation of CMC layer coverage, and new section 2.4 describing estimation of surface coverage by the binders in as-fabricated and calendered electrodes). The editing instructions of *Nature Communications* mention that the Discussion and Conclusions should not be split into sub-sections, thus we abstain from further dividing this part of the manuscript.

Reviewer #3 (Remarks to the Author):

The author develops an accessible approach of staining carboxymethylcellulose (CMC) and styrene butadiene rubber (SBR) binders in graphitic Li-ion electrodes with silver and bromine, enabling detailed electron imaging and precise spectroscopic quantification of the binder domain. Notably, the staining reactions of Br₂ vapour is similar to OsO₄ staining but with significantly lower hazard, as indicated by a 217-times higher short-term safe exposure limit for Br₂ compared with OsO₄. This work can provide a guidance for slurry processing and electrode preparation of graphite anode. Therefore, this work has a chance to be published after major revise:

1. As shown in Figure 1A, the schematic diagram shows that the Ag⁺ can replace the Na⁺ in Na-CMC. It is unscientific, because the Ag⁺ tends to undergo cross-linking reactions with hydroxyl and carboxyl groups rather than substitution reactions.

While we initially also suspected OH to play role in Ag binding to CMC, ATR-IR has shown no differences in the 3200 cm⁻¹ OH band in the material after Ag staining. Recently, we have also attempted Ag⁺ staining of polyvinyl alcohol (PVA) that is rich with OH groups but lacks carboxylate groups, and we found no significant Ag⁺ binding to this polymer after subsequent washing with water (the EDX atomic Ag amount normalized to C+O was 0.0015 in PVA compared to 0.10 in CMC). Furthermore, EDX and XPS analysis of CMC before and after Ag staining showed complete substitution of Ag for Na after silverization. Thus, our experimental evidence suggests that the critical interaction that stabilizes Ag⁺ binding to CMC is the strong complexation by the COO⁻ groups with simultaneous substitution of Na⁺ (that is washed away during the subsequent water rinse), which is highlighted in Fig. 1a. We have now added this information to the first paragraph:

The mechanism of both reactions was confirmed using attenuated-total reflection infrared spectroscopy (ATR-IR) X-ray photoemission spectroscopy (XPS) and energy-dispersive X-ray spectroscopy (EDX), showing diminished aliphatic C=C peaks in brominated SBR, a strong shift of COO⁻ peaks in silverized CMC and a substitution of Ag for Na in the EDX spectrum of Ag-CMC, as discussed in detail in Supporting Notes S1-S2 and Supporting Figures S2-S4.

2. In Figure S4, the EDX and X-ray photoemission spectroscopy (XPS) were used to quantify Br and Ag concentration. However, the measurement results of EDX and XPS are usually

inaccurate when compared with Inductively Coupled Plasma (ICP). It is better to provide the ICP measurement of Br and Ag concentration.

This suggestion motivated us to measure additional electrodes with a varying CMC/SBR ratio (also see our answers to Questions 1.5 and 2.1 above). The primary reason we used EDX for quantification of Br and Ag concentration was that this is the same method used for the following binder mapping in the electrodes, thus it represents the same level of accuracy as the primary method used for binder analysis in the electrodes. While ICP-MS is certainly a more accurate method than EDX, it cannot be applied to spatially map binders in electrodes, being additionally difficult to apply for analysing difficult-to-digest polymers, such as dried SBR or SBR on graphite. Additionally, we have now verified accuracy of Ag and Br EDX quantification on a pure Ag wire and KBr pellet, using the same EDX conditions as the ones used for the Ag and Br concentration analysis in Section 2.1. There was only a very small bias of 1.3% and 3.2 % for Ag and Br quantification, respectively, showing that EDX quantification was already sufficiently accurate. We have now expanded the EDX data to include the 4% relative error due to sample roughness/porosity, giving the complete uncertainty levels at the 95% confidence level within 10% relative. We have updated the Methods with the description of this characterization:

Elemental concentration was calibrated using factory standards provided in the OI Aztec software version 6.1. Systematic uncertainty of the EDX quantification was verified using freshly cleaned Ag wire and dried C-coated KBr pellet with 99.9% (trace metal-basis) and $\geq 99\%$ purity, respectively, giving $1.3 \pm 1.0\%$ and $3.2 \pm 1.5\%$ relative bias for Ag and Br quantification, respectively. Considering $\sim 4\%$ uncertainty due to the roughness/porosity of the graphitic samples,⁴⁷ we estimate the relative combined systematic uncertainty of Ag and Br quantification at the applied conditions and 95% confidence level as $\pm 8.4\%$ and $\pm 10.4\%$, respectively.

The new experiments presented in updated Section 2.1. and Fig. 1 were also now measured on multiple electrodes with changing binder content, providing significantly higher statistical value than the previous ones. The new experiments showed an expected linear trend between Ag or Br concentration and CMC or SBR content in the electrodes. The EDX measurements were shown to be accurate enough to provide required selectivity and sensitivity. Note that all these experiments were measured on at least $5 \times 1.4 \text{ mm}^2$ random spots from large (20 cm^2) homogenized samples of the electrode coatings.

3. As illustrated in Figure 5A, the author claims that the more concentrated CMC solution in the first mixing step might improve the final C45 dispersion in the electrode. What's the specific mechanism? Generally, the concentrated binders solutions are not conducive to dispersion of electrode materials.

We kindly direct you to our discussion in the same section, which we have now extended to clarify the mechanism of CBD de-agglomeration:

We hypothesized that because the amphiphilic CMC electrostatically stabilizes solids dispersion in the slurry by adsorbing onto the graphite and carbon particle surfaces,⁴⁷ a more concentrated CMC solution in the first mixing step might increase CMC saturation on C45 surfaces, improving C45 dispersion in the slurry and in the final electrode.

This is equivalent to acknowledging that CMC acts as a surfactant in the slurry (as evident from the zeta-potential studies in e.g. Ref 47), adsorbing on C45 and graphite surfaces. By increasing the concentration of the surfactant, we increase its saturation on the solid surfaces during mixing, ensuring that it prevents the solids from agglomerating to each other.

4. It is better to provide a optimized concentration of CMC solution in the first mixing step, which is important for the slurry processing in Li-ion battery industry.

As noted, we provide the optimized concentration of CMC solution in the first mixing step as 3% wt.

To reduce CBD agglomeration, we fabricated electrodes of the same dry mass composition, but using a more concentrated CMC solution (increased from 1.5 to 3 wt.%) during an initial mixing step. [...] Fig. 5b shows a BEI image of such modified electrode after bromination, confirming the disappearance of large CBD clusters, which is also evident in the histogram in Fig. 5c.

Furthermore, four-point probe measurements of non-stained electrodes showed that reduced CBD agglomeration decreased mean electronic resistivity in both uncalendared and calendared electrode coatings by 14±2% (Fig. 5d) [...].

We now specify the concentration of CMC in both slurries in the caption of Figure 5. The concentration of CMC solution in the modified slurry mixing protocol is also annotated in the first mixing step on the mixing protocol schematic in Fig. 5a.

5. In Figure 6, when the polymer solution (e.g. PVDF in N-methyl pyrrolidone) is immersed in a non-solvent (e.g. water), the adhesion of polymer slurry on Cu foil will greatly decrease and even detach from the Cu foil surface. This phenomenon will increase the contact resistance of electrode materials and further decrease the electrochemical performance of Li-ion battery. However, the author reports a opposite result by attributing to the reduced pore resistance? I think that it is hard to convince. The author should provide more discussions and comparisons in this section by supplementing experiment.

We did not observe the adhesion problems mentioned above – in fact, we observed a strong increase in adhesion between the electrode coating and the Cu foil after phase inversion in acetone (as mentioned in the text and now rephrased for better clarity):

On the other hand, the electrode immersed in acetone showed a marked binder concentration near the current collector, possibly due to faster precipitation of CMC by acetone with 6× lower viscosity than isopropanol.⁷¹ The greater binder fraction at the current collector allowed bending of the relatively thick electrode without cracking, and it was significantly more adherent to the current collector than the electrodes with the opposite binder distribution (Fig. 6d).

By using binder mapping, this was directly correlated with the increased concentration of the binders (that help with the adhesion) close to the current collector after this treatment (Fig. 6). In the EIS measurements, we also did not observe high-frequency semicircles in Nyquist plots of reference and phase-inverted electrodes that are characteristic to electrode contact resistance, showing that phase inversion did not increase this already-low resistance (see e.g. Landesfeind *et al.*, *J. Electrochem. Soc.*, 163 (2016), DOI: 10.1149/2.1141607jes, or Landesfeind *et al.*, *J. Electrochem. Soc.*, 164 (2017), DOI: 10.1149/2.0131709jes). We have now included this information in the text and provided a supporting figure that compares Nyquist plots of these graphitic electrodes with a NMO/Al cathode measured separately in our group, in which contact resistance is clearly evident:

Also, the process did not change the comparatively low electronic contact resistance of the coatings, as indicated by the absence of a high-frequency semicircle in the Nyquist plots^{3,72} (also see Supporting Figure S19).

Figure S19. Nyquist plots comparing impedance response of: (A) calendered reference- and acetone-phase-inverted graphite/Cu electrodes described in this work and (B) calendered $\text{Na}_{0.7}\text{MnO}_2/\text{Al}$ electrodes measured separately in our group. Insets show high frequency, low impedance regions of the spectra. The measurements were performed at open circuit potential, in symmetric cell configuration, in 10 mM tetrabutylammonium perchlorate in 1:1 wt. EC:DMC, in 100kHz – 10 mHz frequency range. The high-frequency resistive/capacitive semicircle in the $\text{Na}_{0.7}\text{MnO}_2/\text{Al}$ electrode represents the contact resistance between the electrode coating and Al current collector that is characteristic to battery electrodes supported on metallic foils with a native oxide layer.^{23,24} The lack of semicircles in the Nyquist plots of the graphite/Cu electrodes shows that the contact resistance was below the detection limit in both the reference- and acetone-phase-inverted graphite electrodes.

We are unsure what mechanism could lead to the potential delamination concerns mentioned above – we could also not find reports of such an issue in the previous works describing application of phase inversion in electrode manufacturing. These other works (which we also cite in our text) reported similar observation as ours – an increased adhesion strength of the phase-inverted coatings to the current collector. This was previously attributed to e.g. differences in crystallinity of the PVDF binder in phase-inverted coatings or low strain associated with rapid precipitation of binders during phase inversion in the liquid phase, as opposed to the precipitation in a moving drying and shrinking front during conventional slurry drying (see e.g. Harks *et al.*, *J. Power Sources*, 441 (2019), DOI: 10.1016/j.jpowsour.2019.227200; Yang *et al.*, *Adv. Funct. Mater.*, 26 (2016), DOI: 10.1002/adfm.201604229; Wu *et al.*, *ACS Nano*, 15 (2021), DOI: 10.1021/acsnano.1c06491).

Please also note that we attribute lowered pore ionic resistance (which is different to electronic contact resistance) to the changed through-thickness distribution of the binders in the electrodes phase inverted in acetone:

Considering that all electrodes had identical composition, thickness and total porosity, the marked reduction in ionic resistance was a sole consequence of the preferential binder (and porosity) distribution, in agreement with simulations.^{9,73}

6. *As we know, the Br₂ vapours will corrode the Cu foil, CMC and SBR during bromination of electrodes at high temperature, which is is very detrimental to the electrochemical performance of Li-ion battery. Therefore, how to avoid this phenomenon during practical application?*

Thank you for this correct observation. As we explained in our answer to Question 1.3. above and our now-included clarification in Section 2.1, staining is performed on separate electrode pieces for binder mapping only and is not applied to electrode samples subject to other testing (e.g. EIS).

Note that because of the reactivity of Ag⁺_(aq) and Br₂, staining and binder mapping should be applied on separate electrode areas to those used in electrochemical or other testing.

We have also clarified it in Section 2.5 (CBD agglomeration study):

Furthermore, four-point probe measurements of non-stained electrodes showed [...]

And Section 2.6 (phase inversion study):

The greater binder fraction at the current collector allowed bending of the relatively thick electrode without cracking, and it was significantly more adherent to the current collector than the electrodes with the opposite binder distribution (Fig. 6d).

To explore effects on pore ionic resistance, we tested unstained electrodes using electrochemical impedance spectroscopy (EIS) in a non-intercalating electrolyte developed by Landesfeind *et al.*^{3,54}

We have now also included a note in the *Limitations*:

For the same reason, silverization requires impractical detachment of any Cu current collector prior to staining, and the likely alteration of the electrode's electrochemistry by Ag⁺ and Br₂ may restrict its application, for example, in operando studies or on-line manufacturing quality control.

7. The article contains some technical terms and abbreviations (such as EDX, BEI, EsB, etc.). While these are helpful for concise expression, they may pose comprehension barriers for non-specialist readers. It is suggested that the authors provide detailed explanations when these terms first appear.

We have analysed the article again to confirm these terms are explained in their first appearance in the article. EDX, EsB, BEI, CMC, SBR, PVDF, NMP, SEI are first mentioned and explained in the Introduction. XPS, ATR-IR are introduced and explained in Section 2.1 and EIS in Section 2.6 (previous Section 2.5). We have additionally re-explained the most important terms at the beginning of the relevant Results sections (e.g. EDX, BEI are explained again in 2.1 and EsB in 2.3).

8. Although the authors have explained the mechanism of the staining reactions, the descriptions of the optimization processes for some key parameters are not detailed enough. For example, the rationale behind the selection of reaction temperature, time, and bromine vapor concentration in the bromination process has not been fully elaborated. It is suggested that the authors provide more detailed information on these optimization processes to help readers better understand and reproduce the method.

Thank you for this suggestion. We have now extended the Methods section to provide more details on the processing parameters. Please note that the applied conditions of bromination were used without any further optimization – e.g. the provided bromine vapor pressure was not chosen by us, but was simply a consequence of the initial vacuum in the system, temperature of the Br₂ pool and the volume of the Schlenk apparatus:

The electrodes were brominated after vacuum-drying by exposing them to Br₂ vapour in an air-free setup depicted in Figure S1. The reaction was carried out inside a small glass reactor at 50 °C and Br₂ pressure of $5\text{-}8\times 10^{-2}$ mbar for 270 s, followed by outgassing of excess Br₂ under active vacuum ($\sim 2\times 10^{-2}$ mbar). The temperature and time were chosen to ensure no condensation of Br₂ vapour inside the reactor and to facilitate complete reaction with SBR, while minimising the processing time. Br₂ vapour was provided by a small Br₂ pool placed in a separate glass vial immersed in a small water bath (~ 20 °C) and connected to the pre-evacuated and heated reactor ($\sim 2\times 10^{-2}$ mbar, 50 °C) under static vacuum. The bromination conditions were applied without any further optimization. After bromination, samples were kept in N₂-sealed vessel until EDX/BEI analysis to avoid humidity-induced hydrolysis of brominated SBR; however, our later EDX tests also

indicated no significant loss of Br from the electrodes in air for at least two weeks prior to the EDX/BEI analysis.

For silverization, we now also included a note on the use of the surfactant in the experiments for bulk silverization and through-thickness mapping of the electrodes, which was previously missing from the description.

Recently when reproducing the bi-layer electrode silverization (which previously already gave similar results to the bromination of these electrodes) to address Question 3.11 of the same Reviewer, we realized that the original bi-layered silverization experiment was done with an addition of a small amount of non-ionic surfactant (1 mM Triton X-100) in the AgNO₃ solution (the surfactant was not used in any further work, as its solution with AgNO₃ was found to have a limited shelf life). However, recent investigations show that the addition of the surfactant had some benefits for the wetting and accuracy of Ag mapping in the CMC-lean regions of the bi-layered electrodes, due to their specific composition and preparation routine. With this knowledge, we now include surfactant use during the silverization of electrodes presented in the updated Section 2.1. and Fig. 1, as these electrodes required an analogous preparation routine to the bi-layered electrodes. The surfactant was only beneficial in the specific order of sample preparation, staining and then analysis. We have now provided an extended discussion and experimental data on this in the Supporting Note S5 and Supporting Figure S7.

We have now included this information in the main article – in Section 2.2:

The addition of a surfactant (1 mM Triton X-100) to the AgNO₃ staining solution benefited wetting and accuracy of Ag-CMC profiling through the binder-lean region of electrodes, as discussed in the wetting analysis in Supporting Note S6 and Supporting Figure S8.

as well as in the *Methods*:

The electrodes were silverized by immersing them in a crystallizer containing 0.5M AgNO₃ for 3 minutes (depicted in Fig. 1A). For the CMC analysis of homogenized and bi-layered electrodes (*Mechanism and specificity of staining reactions and Staining bi-layered electrodes sections respectively*), the 0.5M AgNO₃ solution was freshly mixed with 1 mM Triton X-100 non-ionic surfactant prior to staining to help with penetration of the solution into all of the electrode, as discussed in detail in Supporting Note S6 and Supporting Figure S8.

Figure S8. A) Schematics of sample preparation, silverization and analysis routine for through-thickness and top-down binder analysis. B) Top-down BEI images of top and internal (bulk) surfaces of an electrode silverized without or with the surfactant. The electrode contained 1.5 wt.% CMC and no SBR. C) EDX Ag quantification in internal- and top surfaces of the same type of electrode silverized without or with the surfactant. D) EDX maps of carbon (brown) and silver (teal) in bi-layered electrodes silverized without and with an addition of 1 mM Triton X-100 non-ionic surfactant to the 0.5M AgNO₃ staining bath. The electrodes contained four times more CMC and SBR in the lower layer compared with the upper layer. E) Ag-L profiles derived from the EDX maps, showing the ratio (Δ) of the Ag-L signal between the binder-rich and -lean layers.

9. While the authors have observed the complex distribution patterns of binders within the electrodes, the in-depth analysis of how these patterns affect the electrochemical performance of the electrodes is not sufficiently thorough. For example, regarding the role of nanoscale binder films during charge-discharge processes and the impact of non-uniform binder distribution on battery cycling stability, it is recommended that the authors conduct a more in-depth investigation in conjunction with the results of electrochemical testing.

We agree fully that these results warrant new detailed investigations on the roles of binder distribution on battery cycling stability and performance, as we have also extensively discussed in the Discussion and Conclusions. Such proper in-depth analysis, however, will necessitate broad testing campaign with carefully designed comparative experiments reproduced on many battery cells to adhere to modern standards in scientific battery community, which would exceed the scope, time and size constraints of this single article.

Because methods for mapping and visualizing binders in battery electrodes are limited, yet urgently sought (as proven by the interest we have received from the academic community and several big battery OEMs) the article focuses on the new binder analysis methods, and already provides several comprehensive application scenarios (Sections 2.3-2.6). However, as a new method we prioritise extensive consideration of the method validation (Sections 2.1-2.2). Together with the additional experiments and new data that was requested and we have now added, the article is now the maximum size for *Nature Communications*. Therefore, battery performance investigations and implications will be described in a subsequent publication.

10. While the article mentions the advantages of CMC and SBR binders over traditional PVDF binders in terms of staining, the comparative analysis with other potential staining methods may not be comprehensive enough. It is suggested that the authors include a comparison with other staining methods in the discussion section or elaborate on the universality of this staining approach.

The only other binder staining technique that we are aware of that has been properly described in the literature is the OsO₄ staining of SBR by Lee *et al*, that we mention in the Introduction:

Most recently, Lee *et al.* described staining of SBR with osmium tetroxide (OsO_4) that could be detected by energy-selective backscattered electron imaging (EsB) or EDX to trace SBR migration in graphite electrodes during drying.^{37,38} The extreme toxicity and volatility of OsO_4 , however, limits this approach to only the most specialized laboratories.

We also discuss how SBR staining by Br_2 compares with this method at the beginning of Section 2.1:

The latter reaction is similar to OsO_4 staining but with significantly lower hazard, as indicated by a 217-times higher short-term safe exposure limit for Br_2 compared with OsO_4 .³⁹

We have now also included discussion of staining other binders at the end of the *Limitations* section:

Also, because the staining methods rely on the presence of carboxylate and aliphatic C=C groups in the binders, any process that affects these groups prior to staining (e.g. drying-induced cross-linking) may lower the fraction of Ag and Br that reacts with the binders (note, however, that this can be also beneficial for quantification of binder cross-linking). Consequently, bromination and silverization are not suitable for staining e.g. PVDF or PTFE binders (that lack these functional groups) but can be suitable for other binders, such as the carboxyl-rich alginates and polyacrylates. However, the accuracy of spatial mapping these two binders in electrodes remains to be fully quantified.

Additionally, in the same section, we have now included details on staining electrodes based on other-than graphite active materials, such as micro- and nano-Si, carbon-coated SiO_x and LiFePO_4 , including new Figure 7 (also attached above, page 12):

First, the staining methods, particularly the liquid-phase silverization, are less applicable to electrodes containing easily-oxidizable materials (e.g. metallic, LiFePO_4 or nano-Si particles), as these can rapidly undergo redox reactions with the staining elements, complicating image interpretation. Micro-Si or materials with an inert, stable surface passivation layer can be, however, suitable. This is shown in Figure 7 that demonstrates successful staining on Si and C-coated SiO_x microparticle-based electrodes but not on the more reactive LiFePO_4 or nano-Si.

11. In presenting key data, such as quantitative analysis of binder distribution, the authors provided EDX and BEI images for only part of the electrodes, which may not fully represent the entire electrode. It is recommended to increase the image coverage to more comprehensively show the overall characteristics of the electrodes. For example, larger -

area images could be provided, or multiple images taken at different positions and stitched together to ensure data integrity and representativeness.

Following this suggestion, we performed a campaign of reproduction tests, and we have now included four additional reproductions of brominated and silverized bi-layered samples in Section 2.2. and updated Figure 2. These tests showed good reproducibility (binder ratio values in individual samples within 3-23% of the averaged values), showing that measuring single 1 mm-cross sections can give representative. Since We have updated the figure and text accordingly with the discussion on the precision and accuracy of through-thickness binder profiling. Following the extended discussion on Br outgassing from the new testing campaign on homogenized electrodes with different SBR:CMC ratio in Section 2.1 and Fig. 1, in Section 2.2. and Figure 2 we now also include comparison of SBR profiling from the absolute EDX Br distribution and the differential (outgassed) Br distribution. Also, having now more cross-sectional BEI data, we have modified the BEI through-thickness quantification routine with a more generalized approach of multilevel Otsu thresholding, giving a more complete picture on the BEI quantification uncertainty.

Figure 2. Sensitivity of binder mapping tested on bi-layered electrodes. a EDX maps of C (brown), Cu (pink) and Ag (turquoise) or Br (cyan) and contrast-equalized backscattered electron images (BEI) of silverized and brominated electrodes with bi-layer binders distribution. **b** EDX- and BEI-derived through-thickness profiles of Ag and Br, showing ratios (Δ) of each signal between the binder-rich and binder-lean layer. **c** EDX-Br profiles of the third brominated sample acquired consecutively at an increasing pixel scanning time (total electron dose), together with Br outgassing profiles obtained by subtracting 5th scan profile from the 1st and 3rd scan profiles. **d** Effects of EDX mapping time (pixel dwell time) on the accuracy of Δ values derived from the absolute Ag and Br signals (points) during the analysis of the 2nd silverized and 3rd brominated samples, respectively, also showing outgassed Br Δ (horizontal bars) calculated between consecutive Br scans. **e** Comparison of Δ values derived using EDX, BEI and Br outgassing in all the samples. Weighted means (columns) were averaged from the data of individual samples (points). The error bars correspond to *Student's t*-95% confidence intervals. All images were acquired at 10 kV and post-processed as described in Supplementary Methods.

While these tests confirmed representativeness of single 1 mm-wide cross section measurements, we also now include in Supporting Figure S18 two matching reproductions of SBR mapping and profiling in separately made, stained and analysed reference and acetone-phase inverted electrodes (the phase inversion study).

Figure S19. EDX maps of Br and C in reference and phase inverted in acetone electrodes after cross-sectioning and Br staining, together with through-thickness Br-L distribution profiles derived from the EDX maps. The profiles exclude the brominated Cu current collector region. The electrodes were separately made and identically processed as the electrodes described in the phase inversion section of the main article.

Reviewer #4 (Remarks to the Author):

In this manuscript, authors suggested a new method that staining carboxymethylcellulose (CMC) and styrene butadiene rubber (SBR) binders in graphitic Li-ion electrodes with silver and bromine, enabling detailed electron imaging and precise spectroscopic quantification of the binder domain. By using this method, the authors could achieve the suppression of binder migration during high-temperature electrode drying, and a 40% decrease in electrode ionic resistance for unprecedented electrode-scale, high-resolution backscattered electron imaging of surprisingly complex binder hierarchies, revealing multiple types of agglomerates and hardly-ever-seen nanoscale binder films. In conclusion, the rich information provided by binder staining can shed new light on fundamental electrode processes and extend optimization routes of negative Li-ion electrodes. Therefore, I suggest this manuscript to be published in journal "Nature Communications " after minor revisions according to the following aspects:

1. The authors conducted special analysis for SE, EDX, BEI. If possible, please provide XPS mapping data for not only elemental analysis but also chemical state analysis.

Thank you for this suggestion.

We have now provided detailed XPS chemical state analysis of C 1s, Ag 3d and Br 3d core levels of CMC and SBR before and after staining, together with survey spectra elemental quantification discussion in Supporting Figs. S3 and S4 and Supporting Note S2. This is now referenced in Section 2.1 in the discussion on the mechanism of staining:

The mechanism of both reactions was confirmed using attenuated-total reflection infrared spectroscopy (ATR-IR), X-ray photoemission spectroscopy (XPS) and energy-dispersive X-ray spectroscopy (EDX), showing diminished aliphatic C=C peaks in brominated SBR, a strong shift of -COO⁻ peaks in silverized CMC and substitution of Ag for Na, as discussed in Supporting Notes 1-2 and Supporting Figures 2-4.

Figure S3. A) Relative amount of Ag and Br (labels) in films of CMC and SBR and powders of graphite (Gr) and C45 carbon conductive additive after silverization and bromination, determined with EDX and XPS. The figure also shows EDX quantification of Ag in silverized sodium alginate (Na-Alg) and sodium polyacrylate (Na-PAA). B) XPS spectra and C) elemental quantification of CMC before and after bromination and silverization. D) XPS spectra and E) elemental quantification of SBR before and after bromination and silverization. The error bars correspond to 95% confidence intervals based on measurements of at least 5 spots for EDX and 3 spots for XPS. In A), the 95% confidence intervals also include systematic quantification uncertainty - 10% relative for EDX and 30% relative assumed for XPS. EDX was acquired at 15 kV beam voltage. Note that the elemental quantification in C) and E) excludes Si found in CMC-Br, SBR-Br and SBR-Ag (1.8 at%, 0.5 at% and 9.0 at%, respectively).

Figure S4. XPS core region analysis of CMC and SBR before and after silverization and bromination. A) C 1s region of CMC, silverized CMC and brominated CMC; B) Ag 3d region of silverized CMC and SBR, together with a spectrum of metallic Ag foil used to create empirical Ag⁰ peak model (the results of fitting Ag foil with asymmetric peak shapes are shown for informative purpose only); C) C 1s analysis of SBR, brominated SBR and silverized SBR; D) Br 3d region of brominated SBR and brominated CMC. For each sample, binding energy was corrected by referencing to the fitted C 1s C-C/C-H peak at 285.0 eV. Dark grey lines represent summed fitting envelopes. Note that due to close proximity of some of the peaks (e.g. C=C and C-H, O-C=O and O-C-O, Ag⁰ and Ag⁺), the peak models and quantitative analysis of C 1s and Ag 3d regions have a significant level of uncertainty. The C 1s and Br 3d spectra were processed with the universal polymer Tougaard background and the Ag 3d spectra with the iterated Shirley background that stabilized the Ag 3d fitting model.

2. In Fig4e, the authors provided graphite surface coverage data to confirm binder distribution. Detailed process to get surface coverage data should be described.

We have included detailed description and algorithm for quantifying binder coverage from EsB images in the Supporting Method S2 and Supporting Code S1. We have referenced the procedure in the caption of Figure 4:

All images were acquired at 1 kV of beam voltage and post-processed as described in the Supporting Method S2.

We have now also included a reference to the Method S2 and Code S1 in the Characterization section of the Methods:

A detailed description of EDX, BEI and EsB image post-processing, including the process for quantification of binder surface coverage, is included in Supporting Method S2 and Supporting Code S1.

3. In this manuscript, the authors only considered graphite as an anode materials. Please provide possibility to use this method to the other electrode materials.

To address this question and similar Questions 1.10 and 3.10, we have now included additional testing results of staining on C/SiO_x, nano- and micro-Si and LiFePO₄-based electrodes in new Figure 7 (also attached above, page 12) and *Limitations* section:

First, the staining methods, particularly the liquid-phase silverization, are less applicable to electrodes containing easily-oxidizable materials (e.g. metallic, LiFePO₄ or nano-Si particles), as these can rapidly undergo redox reactions with the staining elements, complicating image interpretation. Micro-Si or materials with an inert, stable surface passivation layer can be, however, suitable. This is shown in Figure 7 that demonstrates successful staining on Si and C-coated SiO_x microparticle-based electrodes but not on the more reactive LiFePO₄ or nano-Si.

4. In Fig3a, the authors showed images of internal electrode surface exposed by scotch tape peel-off and subsequent silverization, imaged at different electron beam voltages with surface sensitivity. How does the surface sensitivity work with electron beam voltages?

We have now slightly extended the comment on beam voltage-surface sensitivity in Section 2.3 and provided a reference to Goldstein's handbook on electron microscopy, where the dependency of probing depth on the primary electron beam voltage is described in detail:

The images were acquired at low and high magnification (Fig. 3A and 3B, respectively) and gradually decreasing beam voltages, corresponding to decreasing electron-sample penetration depth and increasing surface sensitivity in consecutive images.⁴⁸

The beam voltage-probing depth dependence is also mentioned in the following text:

Consequently, this contrast mostly disappeared at lower beam voltages of 2.5 and 2 kV (shown in Fig. 3a), where the depth probed by backscattered electrons was reduced from a few hundred to a few dozen nanometers (see the simulations in Fig. 3c).

5. The authors claimed that they could reduce 14% of electronic resistivity and decrease 40% of ionic resistance. What is possible way to increase each value?

For electronic resistivity decrease, we have added a comment at the end of the *Correlating slurry mixing...* section

Future staining-assisted optimization could investigate alternative pre-mixing methods of thinner, lower CMC-content slurries, or substituting C45 nanoparticles with carbon nanotubes or graphene for further resistivity decrease.⁶

For pore ionic resistance, we have added a comment at the end of *Phase Inversion* section:

Combining phase inversion with using spherical (instead of flaky) graphite or pore patterning techniques could further improve ionic transport in such electrodes.^{16-19,76}

Besides these modifications suggested by the Reviewers, we have added the following changes to the article:

- Corrected the statement on the origin of commercial electrodes described in line 301 and shown in Fig. 4. In our recent communication with our collaborators, we learned that these electrodes, albeit commercial and industrially-made, were not from an electric vehicle as we were previously told, but originated from LiFUN Technologies (Xinma Industry Zone, Hunan Province, China). This company is a popular manufacturer and commercial supplier of industrially-made electrodes and cells, also used extensively as commercial electrode and cell benchmarks by other research groups. Therefore, we corrected and specified the origin of these electrodes in line 301 to be fully accurate, but there is no change to our findings - our methods working equally well for commercial and laboratory- and/or in-house produced electrodes, and the same binder morphologies in both types of electrodes.

These findings are not limited to laboratory-made electrodes – we also imaged uncycled commercial electrodes from LiFUN Technology (Xinma Industry Zone, Hunan Province, China).

- Modified the figures and captions to adhere to the style of *Nature Communications*
- Added “Si-based” to “graphitic Li-ion electrodes” in the abstract to reflect the updated tests on Si-based electrodes:

Here, we present an accessible approach of staining carboxymethylcellulose (CMC) and styrene butadiene rubber (SBR) binders in graphitic and Si-based Li-ion electrodes with silver and bromine[...]

- Specified the mixer type (centrifugal planetary mixer) used for slurry mixing in Figure 5.
- Corrected the Δ value for the previously-shown Ag-stained bi-layered sample in Section 2.2 from $\Delta = 4.4x$ to $\Delta = 4.8x$. The previous value was calculated based on EDX Ag/(C+O) profiles (adapted during the initial stages of our work). Later, during preparation of the manuscript, we adapted absolute Ag-L (and Br-L) profiles to describe binder cross-sectional distribution, as they accurately represented drop of the element signals at the top and bottom edges of the cross-sections (as visible in Figs 2 and 6). As we realized the originally-presented Ag Δ corresponded to the previous calculation routine, we corrected it to represent Δ based on absolute Ag-L profile and to be consistent with all the other Δ that are based on the absolute Ag-L (or Br-L) signal profiles.

- Substituted discussion on EDX/XPS testing staining specificity on individual battery materials at the beginning of Section 2.1 with a shorter and more streamlined version: EDX and X-ray photoemission spectroscopy (XPS) were used to quantify Br and Ag concentration in the first 1 μm (EDX) and 10 nm (XPS) of the stained materials, showing large amount of Ag in Ag-CMC and of Br in Br-SBR, and little- to no binding of these elements to the other materials, as presented in Supporting Fig. S3.
- Moved the note on the need of delamination of electrode coatings subject to silverization to Section 2.2 (because in the new homogenized electrode analysis in Section 2.1., both brominated and silverized electrodes were delaminated from the current collector before the staining and homogenization).
- Removed the note on contrast-enhancing role of Ag and Br in cross-sectional BEI of bi-layered electrodes from Section 2.2, because such Ag/Br effect was already mentioned in Section 2.1.
- Condensed analysis of SBR migration in a bi-layered electrode made with no SBR in the top layer, previously mentioned at the end of Section 2.2, with the original results of non-homogenized SBR-only electrode analysis (previously shown in Section 2.1). These analysis are now presented together in the discussion on differences in drying-induced migration of SBR and CMC, in Supporting Information and referred to in Section 2.2:
 These average values are within 2.5% and 22% of the expected 4 \times ratio, with more undercounted Δ for the Br-stained SBR due to the larger susceptibility of SBR to drying-induced migration amplified by its weaker adsorption onto graphite⁴⁷, as verified in Supporting Note S5 and Supporting Figure S7.
- Added a brief discussion on the accuracy of pore ionic resistance determination with EIS in symmetric setup to the Methods section, considering the findings of Bieneck *et al* published this month in *J. Electrochem. Soc.* 172 (2025) (DOI: 10.1149/1945-7111/adde17):
 Note that most recently, Bieneck *et al.* showed that the pore ionic resistance determined with symmetric-cell EIS can be strongly influenced by the spatial distribution of the double layer capacitance (C_{dl}) of carbon black, and its contribution to the overall C_{dl} of the electrode.⁷² Based on the mass composition of our electrodes and the BET surface area of the graphite and C45 carbon black (1.47 and 45 m^2/g , respectively), the estimated contribution of C45 to the total C_{dl} of the electrodes was $\sim 25\%$. The analysis of Figure 9 in Bieneck's paper shows that for such C_{dl} contribution, the pore ionic resistance of the

reference electrode (with CBD concentrated close to the separator) determined with EIS may be underestimated by ~12% and that of the acetone-processed electrode (with CBD concentrated close to the current collector) may be overestimated by ~23%, implying that the actual pore ionic resistance difference between these electrodes may be ~57% instead of the apparent 40%.

- Updated the Conflict of Interest statement with a note on a patent application that was filed on the methods described in this paper:

The authors declare the following competing interests: Oxford University Innovation Limited filed provisory patent application 2511810.0 claiming inventorship of Stanislaw P. Zankowski and Patrick S. Grant for aspects of the described method. The application is pending review.

With these changes and additions, we hope we have fully addressed all the Reviewers' questions. Once again, we would like to thank all the Reviewers for your time and providing valuable suggestions, and we are looking forward to your responses.